# The 2018 North Greenland polynya observed by a newly introduced merged optical and passive microwave sea ice concentration dataset

Valentin Ludwig[1], Gunnar Spreen[1], Christian Haas[1,2], Larysa Istomina[1], Frank Kauker[2,3], and Dmitrii Murashkin[1]

[1]Institute for Environmental Physics, University of Bremen, Otto-Hahn-Allee 1, 28359 Bremen, Germany
[2]Alfred-Wegener-Institute for Polar and Marine Research, Am Handelshafen 12, 27570 Bremerhaven, Germany
[3]O.A.Sys - Ocean Atmosphere Systems GmbH, Tewessteg 4, 20249 Hamburg, Germany

**Correspondence:** Valentin Ludwig (vludwig@uni-bremen.de)

**Abstract.** Observations of sea ice concentration are available from satellites year-round and almost weather-independently using passive microwave radiometers at resolutions down to 5 km. Thermal infrared radiometers provide data with a resolution of 1 km, but only under cloud-free conditions. We use the best of the two satellite measurements and merge thermal infrared and passive microwave sea ice concentrations. This yields a merged sea ice concentration product combining the gap-free spatial coverage of the passive microwave sea ice concentrations and the 1 km resolution of the thermal infrared sea ice concentrations. The benefit of the merged product is demonstrated by observations of a polynya which opened north of Greenland in February 2018. We find that the merged sea ice concentration product resolves leads at sea ice concentrations between 60 % and 90 %. They are not resolved by the coarser passive microwave sea ice concentration product. The merged product shows up to 60 % more open water than the passive microwave product during the formation of the polynya. Next, the environmental conditions during the polynya event are analysed. The polynya was caused by unusual southerly winds during which the sea ice drifted northward instead of southward as usual. The daily displacement was 50 % stronger than normal. The polynya was associated with a warm-air intrusion caused by a high-pressure system over the Eurasian Arctic. Surface air temperatures were slightly beneath 0° C and thus more than 20° C higher than normal. Two estimates of thermodynamic sea ice growth yield sea ice thicknesses of 60 and 65 cm at the end of March. This differed from airborne sea ice thickness measurements, indicating that ice growth processes in the polynya are complicated by rafting and ridging. 33 km$^3$ of sea ice were produced thermodynamically.

*Copyright statement.* TEXT

# 1 Introduction

Arctic sea ice influences the climate system by radiating incident heat back into space and by regulating the ocean/atmosphere exchange of heat, humidity and momentum. The fraction of a given area which is covered by sea ice is called sea ice concentration (SIC). SIC is of high relevance for physics, biology and the safety of shipping routes. The summer sea ice retreat observed
since 2007 is a major driver of the Arctic Amplification, the enhanced warming of the Arctic compared to the mid-latitudes (Dai et al., 2019). While the scientific community largely agrees that Arctic Amplification changes the mid-latitude weather patterns, the exact mechanisms and pathways are subject to debate. A comprehensive literature synthesis is given in Vavrus (2018).

Arctic-wide SIC observations are available every second day by spaceborne passive microwave radiometers since 1979 and
daily since 1987 (Tonboe et al., 2016). Passive microwave measurements do not require daylight and are only slightly affected by clouds. Therefore, they can provide data all year and under all weather conditions. The Advanced Microwave Scanning Radiometer (AMSR2) has frequency channels between 6.9 and 89 GHz. The 89 GHz frequency channels are used in this study. The algorithm which we use is the ARTIST (Arctic Radiation and Turbulence Interaction STudy) Sea Ice algorithm (ASI) (Kaleschke et al., 2001; Spreen et al., 2008). The resolution of the 89 GHz channels of AMSR2 goes down to 3 by 5 km in the
instantaneous field of view. Thus, it is possible to retrieve SIC at 3.125 km grid spacing (Beitsch et al., 2014). The resolution of passive microwave sensors ranges from 40–50 km for the 19 and 37 GHz channels available since 1979 (Ivanova et al., 2014; Comiso, 1995; Markus and Cavalieri, 2000) to 5 km for the 89 GHz channels available since 2001 (Kaleschke et al., 2001; Spreen et al., 2008). The spatial and temporal coverage of passive microwave SIC and their year-round availability makes them valuable for climate research. However, the coarse resolution prevents accurate resolution of the sea ice edge, newly formed
polynyas and leads. Polynyas are non-linearly shaped openings in the sea ice (WMO, 1970), leads are linear openings in the sea ice (Marcq and Weiss, 2012; Wernecke and Kaleschke, 2015) which are typically smaller than polynyas

Thermal infrared data as acquired by the Advanced Very High Resolution Radiometer (AVHRR, since 1979), the Visible Infrared Imaging Radiometer Suite (VIIRS, since 2011) and the Moderate Resolution Imaging Spectroradiometer (MODIS, since 1999/2002) offer resolutions of 750 m (VIIRS) and 1 km (MODIS, AVHRR). An algorithm to derive SIC at 1 km reso-
lution from MODIS thermal infrared measurements has been presented and evaluated by Drüe and Heinemann (2004, 2005). Compared to a typical AMSR2 5 by 5 km grid cell, this allows 25 subpixel measurements and thus an enhanced potential to resolve leads.

Leads are not expected to show up as completely open water areas in the thermal infrared data since they refreeze rapidly, especially in winter. However, they still show up as reduced SIC while the sea ice is thin. They are responsible for more than
70 % of the upward ocean-atmosphere heat flux in the central Arctic during winter (Marcq and Weiss, 2012). According to Marcq and Weiss (2012), 1 % of lead area fraction can change the surface air temperature by 3.5° C, hence the thermal infrared SIC is quite sensitive to the presence of leads. In contrast, passive microwave measurements do not resolve narrow leads because of their coarse resolution. Also, 89 GHz measurements are insensitive towards the sea ice thickness for thicknesses above 10 cm (Heygster et al., 2014; Ivanova et al., 2015). While the high spatial resolution of thermal infrared measurements is

a valuable benefit, they are only available in cloud-free locations and thus not suitable if one wants complete spatial coverage as it is needed for long-term climate monitoring.

Synthetic Aperture Radar (SAR) data (Karvonen, 2014; Murashkin et al., 2018) have even higher spatial resolution, e.g. Sentinel-1 A/B with about 90 by 90 m in the Extra Wide swath used over the Arctic Ocean. Further, they penetrate clouds.

If cloud cover is taken into account, there are more SAR data than thermal infrared. However, automated SIC retrieval from SAR measurements is difficult, although attempts have been undertaken, e.g. in Karvonen (2014). Further, the availability and coverage are still limited by the duty cycle and the swath width, so that complete daily Arctic-wide coverage is not guaranteed.

This paper for the first time presents a merged product from AMSR2 passive microwave SIC and MODIS thermal infrared SIC at a spatial resolution of 1 km. This merged product benefits from both the high resolution of the MODIS thermal infrared

data and the spatial coverage of the AMSR2 SIC. A Sentinel-1 SAR-based lead area fraction product (Murashkin et al., 2018), is used for comparison. The benefit of merged SIC towards single-sensor passive microwave or thermal infrared SIC is demonstrated during the formation period of the polynya which was observed between February 14th and March 8th 2018 north of Greenland (Fig. 1).

Polynyas typically last days to weeks and most of them occur regularly (Morales-Maqueda et al., 2004). It can be argued

whether a refrozen polynya which is covered by thin sea ice should be considered as polynya. An argument for referring to it as a polynya is that the heat flux is considerably higher for thin sea ice than for thick sea ice. Also, a refrozen polynya is often visible in SAR images because the sea ice grown thermodynamically under calm weather conditions has a smooth surface. On the other hand, passive microwave measurements retrieve it as fully sea ice-covered as soon as the sea ice is thicker than about 10 cm or covered by frost flowers or snow (Heygster et al., 2014). In this paper, we refer to the polynya as opened as long as

the merged SIC are beneath 100 %.

There are two types of polynyas: sensible and latent heat polynyas. Morales-Maqueda et al. (2004) describe both types of polynyas in detail. We continue with the description of latent heat polynyas since the one we investigate pertains to this type. Latent heat polynyas normally develop close to the coast due to off-shore winds and/or ocean currents which cause divergent sea ice motion. Sea ice is pushed away from the coast and new frazil/grease ice forms. Single latent heat polynyas produce up

to $800 \, \mathrm{km}^3$ per year of sea ice (Tamura and Ohshima, 2011). Heat fluxes are typically between $300 \, \mathrm{Wm}^{-2}$ and $500 \, \mathrm{Wm}^{-2}$ (Haid and Timmermann, 2013; Martin et al., 2004).

Polynyas form the basis for food webs by enabling photosynthesis and provide food for mammals, birds and humans alike (Smith et al., 1990; Morales-Maqueda et al., 2004; Schledermann, 1980). Preußer et al. (2016) report that polynyas between January and March have sizes between 400 and $43{,}600 \, \mathrm{km}^2$. Many polynyas recur annually in the same places (Morales-

Maqueda et al., 2004; Preußer et al., 2016), but the one we investigate does not appear frequently (Fig. 1).

The sea ice north of Greenland is one of the oldest and thickest in the entire Arctic (Vaughan et al., 2013). This sea ice was blown off-shore in the course of days in February 2018, forming a coastal polynya which lasted from February 14th to March 8th and spanned more than $60{,}000 \, \mathrm{km}^2$ at its maximal extent. A very recent study (Moore et al., 2018) uses the ASI-AMSR2 SIC mentioned above to show the polynya. We demonstrate the benefit of our higher-resolution merged SIC product when

describing the formation of the polynya. Moore et al. (2018) identify a sudden stratospheric warming as trigger of the polynya.

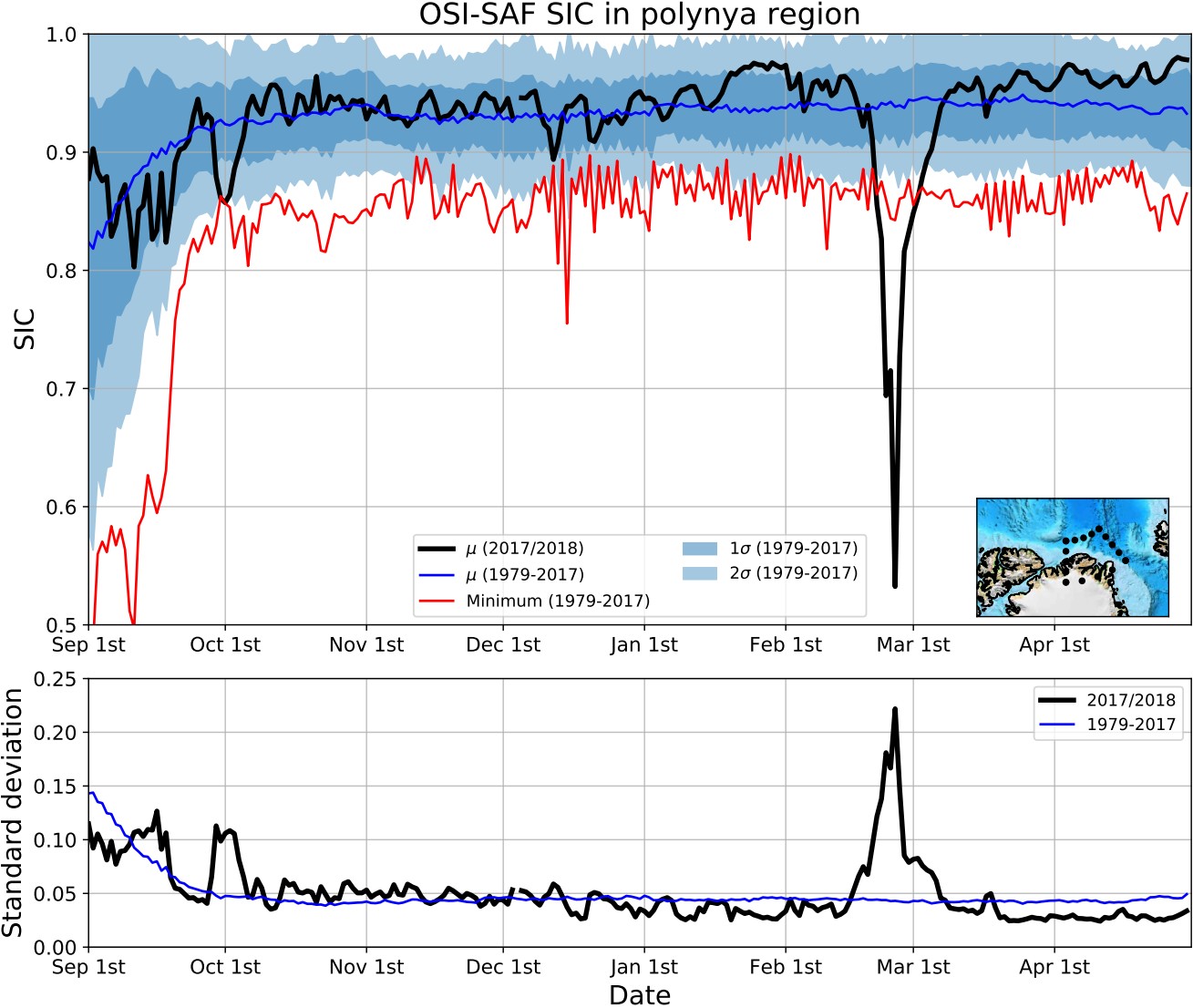

**Figure 1.** Upper panel: Mean OSI-SAF SIC (Lavergne et al., 2019) in the polynya region (indicated by the dashed box on the map in the lower right corner). The black line shows the mean SIC in 2018. The blue line shows the mean SIC between 1979 and 2017. The dark/light shades indicate the 1-/2σ interval, respectively. The red line shows the minimal mean SIC between 1979 and 2017 for each day. Lower panel: Time series of the standard deviation in the polynya region for 2018 (black). The blue line shows the mean of the standard deviations in the polynya region between 1979 and 2017.

We add to their work by investigating sea ice drift data. We conclude by estimating the amount of sea ice which grew in the polynya and the amount of heat released to the atmosphere.

This paper is structured as follows: Sect. 2 describes the data used. Sect. 3 describes the merging procedure and the calculation of the thermal infrared MODIS SIC. Sect. 4 compares the SIC datasets and provides basic information about the polynya itself. Sect. 5 describes the local and large-scale 2 m air temperature, the surface air pressure and the sea ice drift during the opening and refreezing of the polynya. Sect. 6 gives an estimate of the sea ice growth and the heat release in the polynya. Sect. 7 discusses the results. Sect. 8 summarises the results and presents the conclusions. Sect. 9 names directions for future research.

The following questions will be addressed and answered in this paper:

1. Does merging MODIS thermal infrared and AMSR2 passive microwave SIC allow additional insights about the formation of the polynya?

2. Was the polynya opened thermodynamically or dynamically and how unusual were the environmental conditions?

3. How much sea ice grew in the polynya and how much heat was released to the atmosphere?

## 2 Data

This section describes the input data for the merged product (Sect. 3.2) and the data used to investigate reasons and consequences of the polynya formation (Sect. 2.2).

### 2.1 Merging

#### 2.1.1 AMSR2 sea ice concentration

The Global Change Observation Mission-Water Satellite 1 (GCOM-W1) carries the Advanced Microwave Scanning Radiometer 2 (AMSR2, https://suzaku.eorc.jaxa.jp/GCOM_W/w_amsr2/whats_amsr2.html). It has an inclination of 98° and crosses the equator at 1:30 am/pm on its descending/ascending orbit. AMSR2 measures brightness temperatures at six microwave frequencies between 6.9 and 89 GHz. The 89 GHz brightness temperatures have the highest spatial resolution. It is 5 by 5 km in the effective field of view. The ASI algorithm (Kaleschke et al., 2001; Spreen et al., 2008) derives SIC from these brightness temperatures. To get the smallest possible time lag to the MODIS data, we use swath data. Additionally, the swath data have a spatial resolution of 5 by 5 km while the grid spacing of the daily product is 6.25 by 6.25 km. The daily product is publicly available at www.seaice.uni-bremen.de, the swath data have been processed internally.

#### 2.1.2 MODIS data

The Moderate Resolution Imaging Spectroradiometer (MODIS, https://modis.gsfc.nasa.gov/) aboard NASA's Terra/Aqua satellites provides data in the optical and thermal infrared spectrum since 1999 (Terra) respectively 2002 (Aqua). The Aqua satellite's orbit characteristics are similar to those of GCOM-W1. It has the same inclination and flies 4 minutes behind GCOM-W1. The Terra satellite's equator crossing time is shifted by 45 minutes relative to Aqua. We therefore exclusively

use MODIS Aqua and omit MODIS Terra data. The MOD29 ice surface temperature dataset was developed by NASA's Goddard Space Flight Center (Hall and Riggs., 2018) and is distributed by the National Snow and Ice Data Center (NSIDC) at https://nsidc.org/data/MOD29/versions/6. It has a spatial resolution of 1 km. The data are distributed as granules of five minutes length. For cloud screening, we use the MOD35_L2 cloud mask (Ackerman et al., 2017). A pixel is discarded if it is not
labeled as "confident clear" or is over land or is at the coast or is labeled as "cirrus cloud" or "shadow".

## 2.2    Additional data

### 2.2.1    OSI-SAF sea ice concentration

For climatological reference, we use the OSI-450 SIC Climate Data Record product of the European Organisation for the Exploitation of Meteorological Satellites (EUMETSAT) Ocean and Sea Ice Application Facility (OSI-SAF) which is available
from 1979 to 2015 at http://osisaf.met.no/p/ice/ (Lavergne et al., 2019). The data are provided daily since 1987 and every two days before. They are gridded to a Lambert Azimuthal equal area grid, also known as EASE grid 2.0, with a grid spacing of 25 km. For the years 2015–2018, the OSI-401 SIC product is used. The time series of both products is consistent at the transition (Lavergne et al., 2019). For calculating the polynya area time series, we project all data to a north polar stereographic grid with the true latitude at 70° N ("NSIDC grid", https://nsidc.org/data/polar-stereo/ps_grids.html) with 12.5 km grid spacing.
The average of all OSI-SAF SIC between 45° W/81° N and 5° W/85° N in geographic coordinates is used for the polynya area time series in Fig. 1. The polynya region is shown in the inset of Fig. 1. The climatology comprises the years from 1979 to 2017.

### 2.2.2    SAR sea ice concentration

In addition to SIC, sea ice drift, air temperature, and air pressure, lead area fraction is analysed. It is calculated as a fraction of
leads in the area. Binary lead maps are produced by an automatic classification algorithm from Sentinel-1 C-band SAR data at 5.4 GHz (Murashkin et al., 2018). The lead classification algorithm analyses backscatter values and image texture of the surrounding area. Here leads are assumed to be areas of open water of an arbitrary shape. Therefore the polynya is expected to have a high lead area fraction. Sentinel-1 scenes taken in the Extra Wide Swath mode with 40 m pixel size are used. Images taken within every one day are combined in lead maps of the Arctic with 80 m resolution. Then the lead area fraction is
calculated from these binary maps on a 800 m grid. Finally, the data are resampled to the NSIDC grid with 1 km grid spacing for comparison with the other SIC datasets. Sea ice concentration is derived by inverting the lead area fraction:

$$SIC_{LAF} = 1 - LAF, \tag{1}$$

where LAF is the lead area fraction. The product is called SAR SIC.

### 2.2.3 Sentinel mosaics

Since fall 2014, the Technical University of Denmark has produced Near Real Time mosaics of Sentinel-1 SAR data as they become available to the Copernicus Marine Environment Monitoring Service (CMEMS). The mosaics cover most of the potentially sea ice covered areas of the Northern and Southern hemispheres respectively. They consist of geometrically and radiometrically corrected data from Extra Wide Swath and Interferometric Wide Swath modes of both Sentinel-1A and Sentinel-1B. The radiometric correction includes a correction for the average incidence angle dependence of the sea ice backscatter. The full mosaics are available at http://www.seaice.dk.

### 2.2.4 Sea ice drift, air temperature and air pressure

The OSI-405 low resolution sea ice drift product by EUMETSAT OSI-SAF (Lavergne et al., 2010) is used in this study. It has a grid spacing of 62.5 km, a temporal resolution of two days and is projected to the NSIDC grid. Sea ice motion is first derived separately from ASCAT (Advanced Scatterometer) C-band backscatter, AMSR-E/AMSR2 37 GHz, SSM/I 85 GHz and SSMI/S 91 GHz brightness temperatures. Then, the single-sensor sea ice drift vectors are merged by an optimal interpolation scheme. A comparison to other sea ice drift datasets is given in Sumata et al. (2014).

We use station 2 m temperature data from the weather station at Cape Morris Jesup operated by the Danish Meteorological Institute. They are sampled in three-hour intervals until 2015 and hourly since 2016. We average the values daily. Additionally, we use surface air pressure and 2 m air temperature at a spatial/temporal resolution of 0.25 degrees/one day. They are obtained from the ERA5 reanalysis (https://confluence.ecmwf.int/display/CKB/ERA5+data+documentation,Copernicus Climate Change Service,2015). The ERA5 reanalysis is run at the European Centre for Medium-Range Weather Forecasts (ECMWF). It is the fifth generation of reanalyses from ECMWF. Hourly reanalysis data of 2 m air temperature and 10 m wind are available in near-real time at a spatial resolution of 31 km (Hersbach and Dee, 2016).

### 2.2.5 Sea ice growth from freezing degree days

To estimate thermodynamic sea ice growth in the polynya, we employ an empirical equation described by Lebedev (1938):

$$SIT[cm] = 1.33 * FDD^{0.58} \tag{2}$$

where SIT is the sea ice thickness in cm and FDD are freezing degree days. Freezing degree days are the sum of air temperatures above/below freezing over a given time, where air temperatures below/above $0°$ count positively/negatively:

$$FDD = \sum_{i=1}^{i=n}[-1 * T_{air}] \tag{3}$$

where i is the number of days and $T_{air}$ is the daily mean air temperature in $°$ C, in our case the ERA5 2 m air temperature. We will compare sea ice thickness from different sources. For a consistent comparison despite the very different grids, we introduce a grid-independent criterion for the polynya region: We consider only those grid cells where the sea ice concentration was below 50 % at least once during the polynya event. For the freezing degree days, we use the ERA5 sea ice concentration. In addition

to sea ice thickness, we calculate the sea ice volume produced by thermodynamic growth. For this, we multiply the sea ice thickness with the fixed area of grid cells which were at least once beneath 50 % SIC while the polynya was open.

### 2.2.6 Passive microwave sea ice thickness

Sea ice thickness up to 50 cm can be derived from 1.4 GHz passive microwave measurements (Huntemann et al., 2014; Paţilea et al., 2019). We use the combined sea ice thickness product of the Soil Moisture and Ocean Salinity (SMOS) and Soil Moisture Active/Passive (SMAP) radiometers to evaluate the sea ice growth from the freezing degree days. The product is disseminated by the University of Bremen at https://seaice.uni-bremen.de. It comprises both dynamic and thermodynamic growth. We need to ensure that we consider only those grid cells with thermodynamic sea ice growth. Therefore, we apply the same criterion as described in Sect. 2.2.5. We select only grid cells where the passive microwave sea ice concentration was beneath 50 % at least once during the polynya.

### 2.2.7 NAOSIM model

The North Atlantic Arctic Ocean Sea Ice Model (NAOSIM, Kauker et al., 2003) has been used to calculate the sea ice growth and the vertical heat fluxes during the polynya event. We want to avoid interpolating from the model grid to the NSIDC grid. For a consistent selection of grid cells with thermodynamic sea ice growth, we select the model grid cells which had beneath 50 % SIC at least once during the polynya event, as described in Sect. 2.2.5, and perform the calculations on the model grid which is described in the next paragraph.

NAOSIM's ocean model is derived from version 2 of the Modular Ocean Model (MOM-2) of the Geophysical Fluid Dynamics Laboratory (GFDL). The version of NAOSIM used here has a horizontal grid spacing of 0.25° on a rotated spherical grid. The rotation maps the $30°W$ meridian onto the equator and the North Pole onto $0°E$. In the vertical it resolves 30 levels, their spacing increasing with depth. The ocean model is coupled to a sea ice model with viscous-plastic rheology (Hibler, 1979). The thermodynamics are formulated as a zero-layer model following Semtner (1976). Freezing and melting are calculated by solving the energy budget equation for a single sea ice layer with a snow layer and an ocean mixed layer according to Parkinson et al. (1979). In contrast to the original formulation the energy flux through the sea ice is calculated by a PDF for the distribution of sea ice thickness based on airborne electromagnetic measurements (Castro-Morales et al., 2014). The sea ice model's prognostic variables are sea ice thickness, sea ice concentration, and snow depth. Sea ice drift is calculated diagnostically from the momentum balance. All quantities are mean quantities over a grid box. When atmospheric temperatures are below the freezing point, precipitation is added to the snow mass. The snow layer is advected jointly with the sea ice layer. The surface heat flux is calculated using prescribed atmospheric data and sea surface temperature predicted by the ocean model. The sea ice model is formulated on the ocean model grid and uses the same time step. The models are coupled following the procedure devised by Hibler and Bryan (1987). At the open boundary near 50° N the barotropic oceanic transport is prescribed from a coarser resolution version of the model that covers the whole Atlantic north of 20° S Köberle and Gerdes (2003).

In contrast to the version described by Kauker et al. (2003), the present version uses a modified atmospheric forcing data set consisting of 10m-wind velocity, 2m-air temperature, 2m-specific humidity, total precipitation, and downward solar and

thermal radiation. For the period from 1979 to 2010 the forcing is taken from the National Center for Environmental Prediction (NCEP) Climate Forecast System Reanalysis (NCEP-CFSR) (Saha et al., 2010) and for the period from 2011 onwards from the NCEP Climate Forecast System version 2 (CFSv2) (Saha et al., 2014).

The initial state of January 1 1980 is taken from a hindcast from January 1 1948 to December 31 1979. As in Kauker et al.
5  (2003) this hindcast run was forced by the NCEP/NCAR reanalyses (Kalnay et al., 1996) and, in turn, initialized from the Polar Science Center Hydrographic Climatology (Steele et al., 2001) (ocean temperature and salinity), zero snow depth, and a constant sea ice thickness of 2 m with 100 % sea ice concentration where the air temperature is below the freezing temperature of the ocean's top layer.

Recently the model parameters were optimised with the help of a genetic algorithm. One set of optimised parameters out
10  of eleven independent optimisations (set number three) is used for the simulation employed in this study starting in 1980. For details on the optimisation procedure and results of the optimisation, i.e. the quality of the simulation we refer to Sumata et al. (2019a) and for a detailed analysis of the independent eleven optimised parameter sets we refer to Sumata et al. (2019b).

## 3 Methods

### 3.1 MODIS sea ice concentration

To calculate the MODIS SIC, we adapt the approach used in Drüe and Heinemann (2004). They interpolate linearly between the ice surface temperature of a fully sea ice-covered pixel (sea ice tiepoint $IST_I$) and that of a fully water-covered pixel (water tiepoint $IST_W$):

$$SIC = 1 - \frac{IST_{obs} - IST_I}{IST_W - IST_I},\tag{4}$$

where $IST_{obs}$ is the observed ice surface temperature. $IST_W$ is set to -1.8° C, the freezing point of sea water. For $IST_I$, the local variability of the ice surface temperature has to be taken into account. MODIS granules normally have 2030 by 1054 pixels. We crop them so that the dimensions are divisible by 48. Then a box of 48 by 48 pixels, called one cell, is taken. The cell is divided into three by three subcells of 16 by 16 pixels. The 25th percentile of each subcell is selected as preliminary sea ice tiepoint. The choice of the percentile does not have significant impact on the final tiepoint (Lindsay and Rothrock, 1995). The sea ice tiepoint for each pixel is then expressed as linear function with two variables:

$$IST_I(x,y) = ax + by + c,\tag{5}$$

where $a$, $b$ and $c$ are determined by bilinear regression and x and y are the x/y coordinates of the respective pixel within the cell. Subcells are discarded if more than 70 % of the pixels are masked out by the cloud mask. Cells are discarded if they contain less than five valid subcells. Then, the next 48 by 48 pixel box is processed. So far, we strictly follow the approach by Drüe and Heinemann (2004). Then, in difference to them, we shift the box by only one pixel at a time and repeat the bilinear regression before shifting the box by the next pixel. Thus, there are 48 possible $IST_I$ values per pixel. The mean of those values is selected as $IST_I$. Subsequently, each granule is projected to the NSIDC grid at 1 km grid spacing to be merged with the next closest AMSR2 swath. Of one MODIS granule, there were on average 36 % cloud-free pixels. When considering all granules of one day, 80 % of the pixels were covered at least once.

### 3.2 Merging

For each MODIS granule, the AMSR2 swath with the closest acquisition time is selected. On average, 8 MODIS/AMSR2 matching overflights are available per day. The time lag between MODIS and AMSR2 is normally between three and eight minutes since both satellites, Aqua and GCOM-W1, fly in the A-Train satellite constellation. The A-Train is a suite of satellites which follow each other closely on the same orbit. It was designed to obtain near-simultaneous Earth observation data from different measurements.

AMSR2 SIC are given as half-orbits starting either at the North Pole (descending orbit) or at the South Pole (ascending orbit). For a descending orbit, we take the time of the first measurement as acquisition time. For an ascending orbit, we take the time of the last measurement as acquisition time. For the MODIS SIC, we take the starting time of that granule as acquisition time. We use the so-found MODIS/AMSR2 pair if it has at least 10 % of cloud-free overlap. For the merging, we split the

MODIS data in boxes of 5 by 5 km, which roughly corresponds to one AMSR2 footprint. The MODIS and AMSR2 SIC in this 5 by 5 km box are called $SIC_{MODIS,5km}$ and $SIC_{AMSR2,5km}$, respectively. Now, we calculate the difference between the two datasets, $\Delta_{SIC,5km}$, for each box:

$$\Delta_{SIC,5km} = SIC_{AMSR2,5km} - SIC_{MODIS,5km}. \tag{6}$$

$\Delta_{SIC,5km}$ is now added to the MODIS SIC as shown in Eq. (7). This way, we preserve the mean of the AMSR2 SIC in this 5 by 5 km box. In a last step, we use the AMSR2 data where no MODIS data are available:

$$SIC_{merged_{i,j}} = \begin{cases} SIC_{MODIS,5km_{i,j}} + \Delta_{SIC,5km}, & \text{if } SIC_{MODIS,5km_{i,j}} \text{ available} \\ SIC_{AMSR2,5km_{i,j}}, & \text{if } SIC_{MODIS,5km_{i,j}} \text{ not available} \end{cases} \tag{7}$$

where the indices $i, j$ denote the position within the 5 km box. To get a smooth field, the box is then shifted by 1 km and the procedure is repeated, before the box is again shifted by 1 km. This way, each pixel is covered 25 times. The mean for each pixel is selected as merged SIC value. This procedure preserves the AMSR2 mean within the 5 by 5 km box, so that there are no sudden in-/decreases of SIC if no MODIS pixel is available. A similar procedure has been applied by Gao et al. (2010). If the AMSR2 SIC is 100 %, the merged SIC at single pixels can be above 100 %. We tolerate this because we want to preserve the mean SIC from AMSR2. Merged SIC above 100 % are set to 100 % in the end.

### 3.3 Open water extent

We want to show the benefits of the higher resolution of the merged SIC compared to the AMSR2 SIC. The mean SIC for both datasets is identical by definition. However, the higher resolution of the merged product results in sharper gradients, e.g. at the edges of leads. To show this effect, we calculate the open water extent for both datasets. It is defined as the area covered by all pixels which has at least 15 % open water. Due to its higher spatial resolution, the merged SIC are expected to have a higher open water extent than the AMSR2 SIC. For meaningful comparison, we consider only those data points where cloud-free MODIS data are available for the merging. Also, we constrain our analysis to scenes when at least 50 % of the pixels are cloud-free measurements. The open water extent is normalised by dividing it by the number of cloud-free pixels.

### 3.4 Airborne ice thickness profiles

We use data of an airborne electromagnetic (AEM) ice thickness survey carried out over the southeastern region of the refrozen polynya on March 30 and 31, 2018, i.e. roughly five weeks after the polynya had begun to refreeze. Surveys were carried out with a DC-3/Basler BT67 aircraft (Haas et al., 2010), and were processed as described by Haas et al. (2009). AEM data have an accuracy of $\pm$ 0.1 m over level ice but can underestimate the thickness of pressure ridge keels by up to 50 % due to the large footprint of the AEM measurement of up to 45 m over which an average ice thickness estimate is retrieved. Accuracy was confirmed by a sufficiently large number of small open leads with ice thickness of zero meters. AEM measurement obtain the total, ice plus snow thickness. Visual observations showed that the snow on the young first-year ice of the polynya was less than 0.05 m thick and can be neglected for the purpose of this study. All measurements over small patches of multiyear ice

embedded in the polynya have been removed from the data set. The results therefore represent the thickness of 5 weeks old first-year ice in those specific environmental conditions.

## 4  Sea ice concentration

This section first compares the 2018 SIC in the polynya region to that of the entire satellite period (1979–today) in Sect. 4.1.
Afterwards the advantage of the merged SIC towards the other, single-sensor products is discussed and demonstrated in Sect. 4.2. Finally, the temporal evolution of the polynya is described in Sect. 4.3.

### 4.1  Climatological context

Figure 1 puts the polynya into a climatological context by comparing it to the OSI-SAF Climate Data Record which goes back to 1979 (Lavergne et al., 2019). Since the Climate Data Record is only available until 2015, we use the operational OSI-SAF SIC product after 2015. The products are temporally consistent at the transition. We show the 1979–2017 mean SIC in the polynya region (box on the inset map) for each day between September 1st and April 30th. Normally, the mean SIC in the region north of Greenland is around 95 %, with a standard deviation of 3 % after the freeze-up period in September and October. The climatological mean and the standard deviation do not change much between the beginning of November and the end of April. Except for a 10 % drop during the early freeze-up at the end of September, the 2017/2018 SIC stayed within one standard deviation of the climatological mean until mid January. There was a two-week period of SIC above the climatological mean in the second half of January. In mid February, the polynya started opening rapidly. The mean SIC was at its minimum at February 26th, when it was close to 70 %. Previously, the lowest mean SIC at any day between October 1st and April 30th was 79 %. The time series of the minimal SIC shows that there were other periods during which the mean SIC was outside of the $2\sigma$ interval in particular years, for example once in mid December 1986, once in early January 1984 and once in late March 1983 (single years not shown in Fig. 1). However, none of them reached the low extent of the 2017/2018 winter season. We also investigated the homogeneity of the sea ice cover by calculating the mean spatial standard deviation in the polynya region (Fig. 1, lower panel). It was above 20 % in 2018 while it is normally close to 5 %. This underlines how strongly the normally homogeneously distributed sea ice cover north of Greenland broke up during this exceptional event.

### 4.2  Sea ice concentration comparison

The advantage of high-resolution SIC datasets and the differences between the single-sensor datasets are illustrated in this section. Figure 2 shows the AMSR2 SIC, MODIS SIC, merged SIC and SAR SIC in comparison (Fig. 2a–d)). West of the polynya, the MODIS SIC are lower than the AMSR2 SIC. The merged SIC preserve the AMSR2 mean and are thus higher than the MODIS SIC and spatially continuous if there are no MODIS data available. The benefit of including the MODIS data can be seen when looking at the leads which open west of the polynya: They are much more clearly resolved in the merged product. This is illustrated in the cumulative frequency distribution in Fig. 2e) in more detail. The SAR SIC are the only product which shows 0 % SIC as they have the finest spatial resolution and are based on a binary product. Additionally, a lead covered

by very thin, smooth sea ice would still be classified as "open water". These leads show up as reduced SIC of around 20 % in MODIS SIC. The broader leads are also resolved by AMSR2 and show up as SIC between 70 and 80 %. AMSR2 retrieves only few values in the range between 40 and 60 %. The higher amount of SIC between 60 and 80 % is where the merged SIC resolves leads which are too narrow to be retrieved by AMSR2. Over the polynya region, the MODIS SIC and the SAR SIC

are higher than the AMSR2 SIC. While AMSR2 retrieves 0 % SIC at the onshore and 20 % at the offshore side of the polynya, MODIS retrieves 40 % SIC at the onshore and 80 % at the offshore side of the polynya. The gradient occurs because the newly formed sea ice is advected away from the coast and pushed towards the northeastern boundary of the polynya. New sea ice forms and piles up at the offshore side of the polynya. Generally, the impact of thin sea ice on the different products can be described as follows: In the very early growth phase, the SAR SIC are close to 0 % as long as the sea ice is smooth. When

the smooth sea ice cover breaks up, the backscatter starts to increase and the SAR SIC increases. Additionally, the algorithm was trained with small leads which have a flat surface (Murashkin et al., 2018). In the polynya area, which is larger, the water surface can be rougher and would therefore not be classified as lead. The MODIS SIC are low during the early growth phase, but not 0 % because the surface air temperature is slightly below the freezing point as soon as there is a very thin layer of sea ice. Their sensitivity to sea ice thickness decreases as the sea ice thickness increases. The AMSR2 SIC are sensitive to sea ice

thicknesses up to 10 cm (Heygster et al., 2014). The merged SIC are less sensitive towards sea ice thickness than the MODIS SIC because they are tuned to preserve the AMSR2 SIC mean. However, because they also include the MODIS information, they still have some sensitivity towards sea ice thickness above 10 cm. The different sensitivities towards sea ice thickness are further illustrated in the time series of the mean SIC in Fig. 3. Note that the mean of the AMSR2 SIC is not shown because it is equal to that of the merged SIC by definition (Section 3.2). The MODIS SIC are lower than the merged SIC while the polynya

breaks up and after it has frozen over. This is because they are more sensitive to the sea ice thickness and thin sea ice is shown as reduced sea ice concentration. During the peak of the polynya area, they are larger than the merged SIC. Here, they are more sensitive to freshly grown sea ice. Also, they are more sensitive to small sea ice surface temperature variations because the range between the sea ice tiepoint and the water tiepoint gets very small. The SAR SIC are also larger than the AMSR2 SIC during the peak of the polynya area. The reason is again that they more sensitive to freshly grown sea ice than the merged SIC.

While recently formed sea ice is retrieved as low SIC by the merged SIC, it increases the backscatter as soon as it breaks up. Due to the drift within the polynya, it is expected that the sea ice surface is not smooth, but breaks up quickly. Additionally, as mentioned above, the algorithm was not trained to classify rough surfaces as water. To further demonstrate the benefit of the higher resolution of the merged product compared to the AMSR2 SIC, we show the open water extent in Figure 4. Consider a typical AMSR2 grid cell of 5 by 5 km with an AMSR2 open water fraction of 10 %. Our 1 km resolution dataset allows the

retrieval of 25 different open water fractions in this grid cell whose mean would still amount to 10 %. However, single cells may have an open water fraction above 15 %, so that the open water extent is expected to be higher for a dataset with higher resolution. A comparison of the time series shows that the difference between the two datasets is small during most of the time. It is 2-3 % while the polynya is open and close to 1 % after it has been closed. The benefit is more apparent when comparing the datasets relative to each other, as shown in the lower panel of Fig. 4. While the polynya is opened, the open water extent of

the merged product is 10–60 % higher.

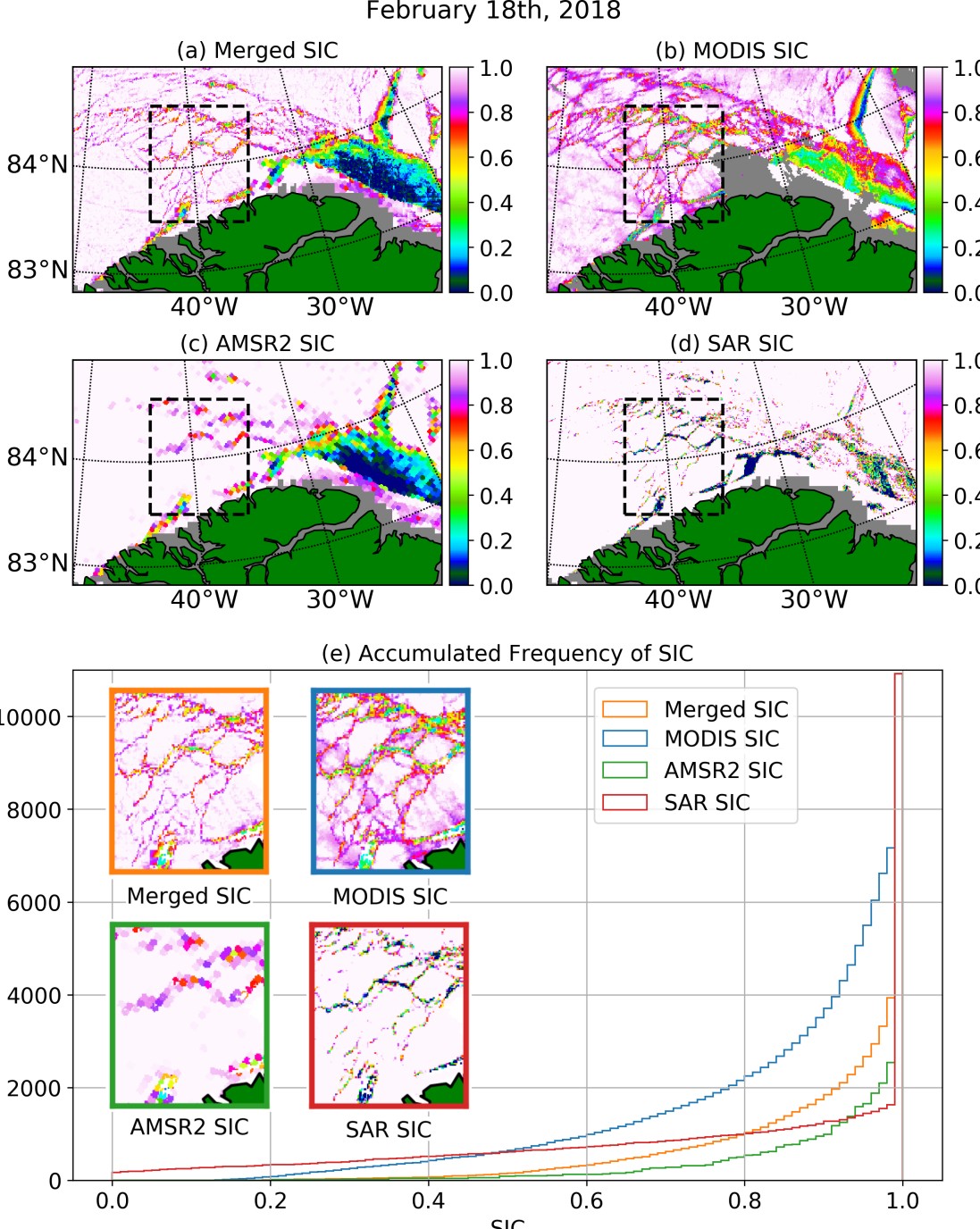

**Figure 2.** SIC on February 18th, 2018 observed with (a) merged, (b) MODIS, (c) AMSR2 and (d) SAR SIC. The acquisition times for MODIS/AMSR2 were 11:45/11:39 am UTC, respectively. The black dashed box in (a)–(d) marks the region used in Fig. 2e). (e) Cumulative histograms for the four datasets. The insets show the SIC distribution for the single datasets. Data points where one of the products was not available were discarded.

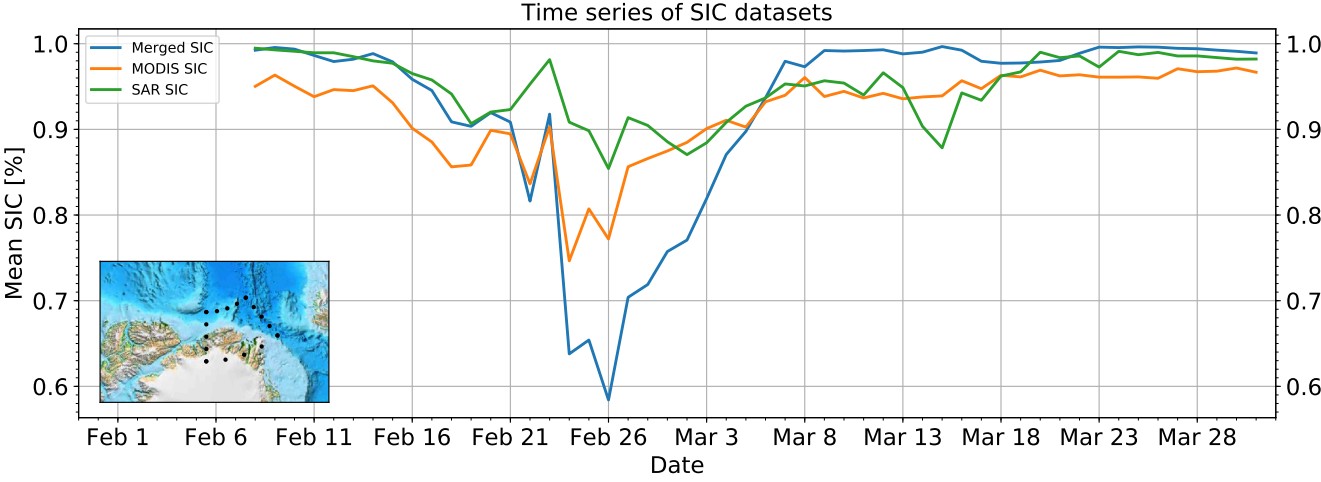

**Figure 3.** Time series of the mean merged, MODIS and SAR SIC in the polynya region. The AMSR2 SIC are not shown because their mean is equal to the mean of the merged SIC by definition. Only points where all datasets were available have been considered. The map in the lower left shows the region which was considered for the time series as dotted polygon.

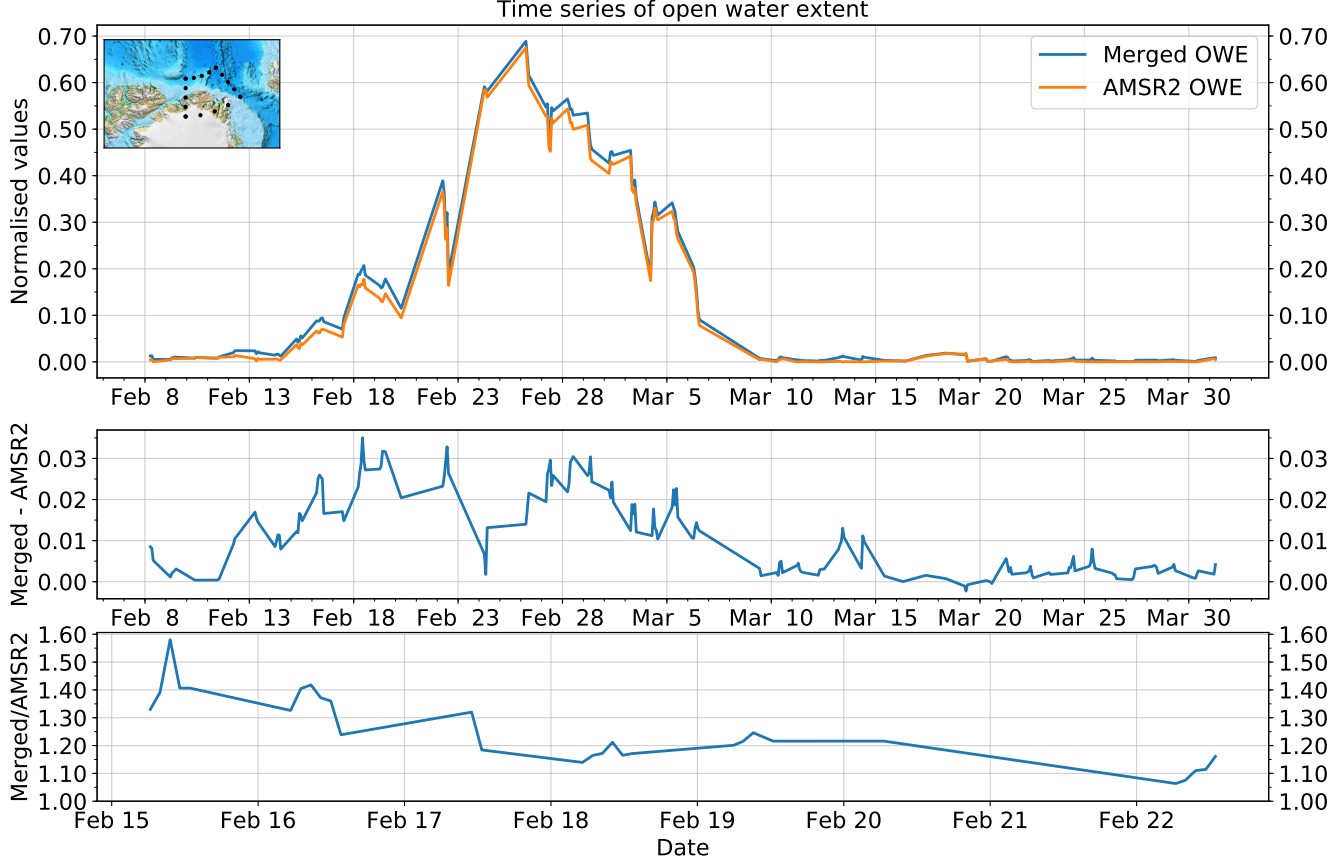

**Figure 4.** Time series of the merged and AMSR2 open water extent. The upper panel shows the absolute value of the open water fraction for both datasets. The middle panel shows the difference of the merged and the AMSR2 open water fraction. The lower panel shows the quotient of the two datasets during the opening phase. Mind the different time span of the lower panel. Only points where MODIS data were available were considered.

## 4.3 Polynya development

Having shown that the polynya was unprecedented in magnitude and having demonstrated the benefit of our merged SIC product, we now focus on describing the temporal and spatial development of the polynya during the opening and refreezing. For this, we show maps before, during and after the polynya event in Fig. 5a)–h), as well as a time series of the open water area from the merged product (Fig. 5i). First leads are already visible on February 8th, six days before the polynya actually starts to open. Also, the shear zone parallel to the coast where the polynya will break up later is already visible. This demonstrates the benefit of the merged product towards the AMSR2 SIC, which would be too coarse to resolve these leads, as seen in Fig. 2. Starting on February 14th, the polynya area increases steadily until February 22nd, when it already spans $30{,}000\,\mathrm{km^2}$. The polynya area decreases on February 22nd and 24th. Apart from this, the polynya area increases strongly until it reaches its maximum extent on February 26th, when it spanned more than $60{,}000\,\mathrm{km^2}$ (Fig. 5). Afterwards, the area decreases almost linearly with time until the now refrozen polynya is not identified as open water any more on March 8th. Note that the area of the opening is still visible as dark/new ice in the Sentinel-1 mosaic.

There are areas (Fig. 5a,g) where leads and $100\,\%$ SIC are directly next to each other. This happens when there are no MODIS SIC available for the merging. In this case, the merged SIC are equal to the AMSR2 SIC which show $100\,\%$ SIC for sea ice thicker than $10\,\mathrm{cm}$.

In the next section, we investigate the driving mechanism behind the polynya and the environmental conditions throughout the event.

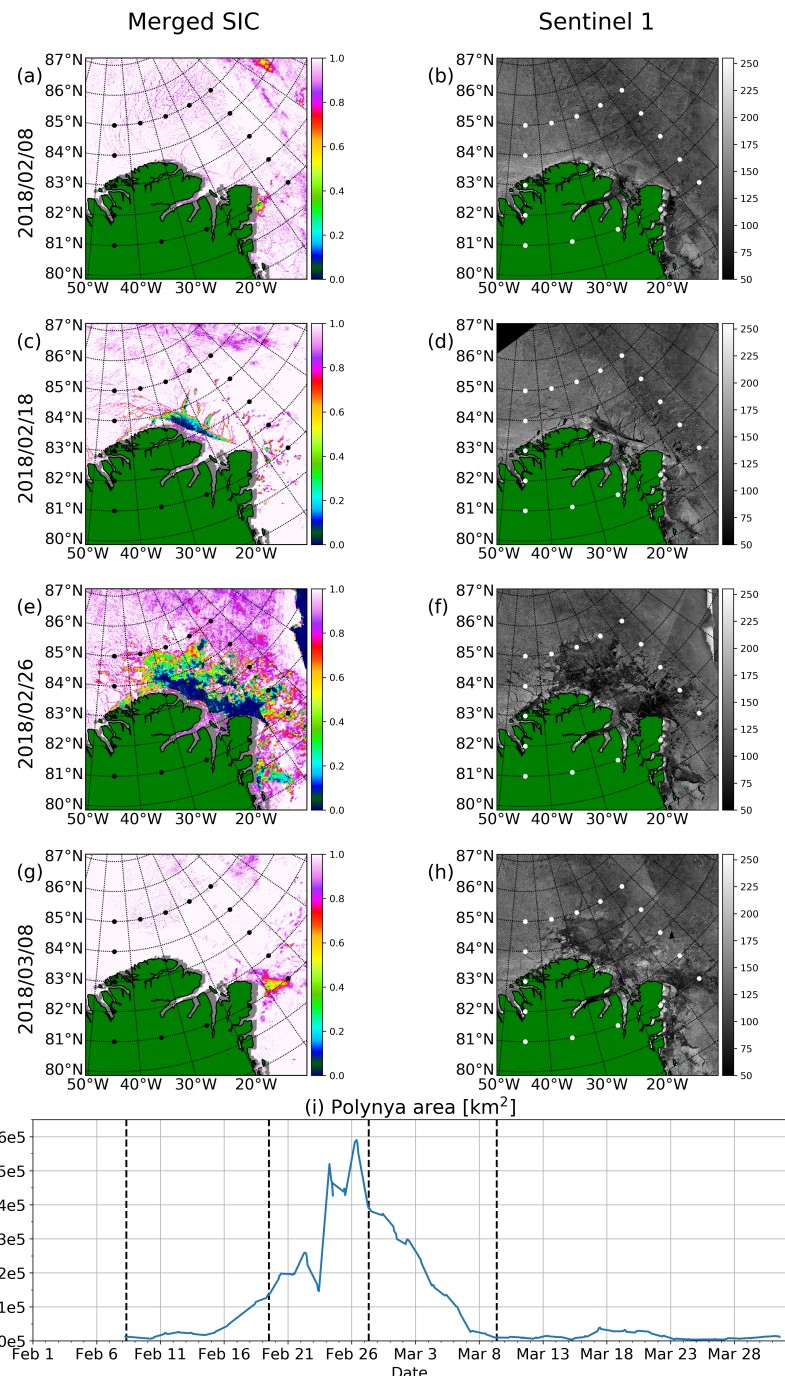

**Figure 5.** (a), (c), (e) and (g): Merged SIC before, at the beginning, at the maximum and after the refreezing of the polynya. The MODIS/AMSR2 pairs were acquired at 07:50/07:44 on February 8th, at 11:45/11:39 on February 18th, at 09:15/09:11 on February 26th and at 11:30/11:28 on March 8th. All times are a.m. and UTC. (b), (d), (f) and (h): Corresponding Sentinel-1A/B daily mosaics. (i): Time series of the polynya area. The polynya area is calculated as sum of the open water fraction (1 - merged SIC) in the map area, multiplied with the respective grid cell size. All available granules are shown. The acquisition times of (a), (c), (e) and (g) are marked by the vertical dashed lines.

## 5 Environmental conditions

There are two possible reasons for the polynya: The sea ice could have drifted away, which would be typical for a latent heat polynya or it could have melted, which would be typical for a sensible heat polynya. This section describes and analyses the 2 m air temperature and surface air pressure (subsection 5.1) and the sea ice drift pattern (subsection 5.2) associated with the polynya.

### 5.1  Air temperature and surface air pressure

Local air temperatures (Fig. 6) at the autonomous weather station in Cape Morris Jesup in 2018 were above the 2010–2017 average from February 15 to March 8. This is in line with the breaking up and refreezing of the polynya. The air temperature increased rapidly at the beginning of the polynya period. During the formation of the polynya, the air temperature varied by more than $10°$ C from day to day and crossed the freezing point on nine out of ten days between February 16th and 25th. The air temperature decreased as soon as the polynya started to refreeze and reached the average value on March 8th. Above-average air temperatures during this time of the year have occurred before, for example in 2011 and 2013. However, those lasted only up to five days and not ten days like during the event studied here. On a larger spatial scale, Fig. 7 and Fig. 8 show the air temperature and surface air pressure distribution during the formation phase (February 22nd to 26th) and the refreezing phase (March 2nd to 4th) of the polynya. During the formation, the air temperature was up to $20°$ C and more than two standard deviations above the average in the polynya region. This was not only a local phenomenon, but associated with a warm-air intrusion from the Atlantic Ocean which caused anomalously high air temperatures until beyond the North Pole (Fig. 7a). The surface air pressure distribution completes the picture: There was a high-pressure system over the Barents and Kara Sea which persisted until February 26th. The surface air pressure was 30–40 hPa above average, which is more than two standard deviations (Fig. 7f). This is the period when the polynya opening rate increased (Fig. 5i). The high-pressure system caused northward winds over the Greenland Sea which contributed to the opening of the polynya. Furthermore, it caused the advection of warm air from the mid-latitudes towards the Arctic region. Ten days later, the atmospheric state had changed substantially (Fig. 8). The air temperature dropped down to the mean of the previous year. The surface air pressure was high over the Central Arctic and lower over the Eurasian Arctic. This caused southward winds which contributed to the closing of the polynya, together with the Transpolar Drift. Although the air temperature was far above average, it was not high enough to explain the polynya. It only crossed the freezing point for some hours, but can not have melted the thick multiyear ice north of Greenland. We conclude that the sea ice must have been broken up by sea ice drift. This is consistent with the study of Moore et al. (2018). They found that the thermodynamic sea ice production was always positive, while the sea ice motion caused the net loss of sea ice. The warm-air intrusion between February 13th and (Fig. 6 and March 3rd 7) contributed to maintaining the polynya open. The next section describes the sea ice drift pattern throughout the polynya event.

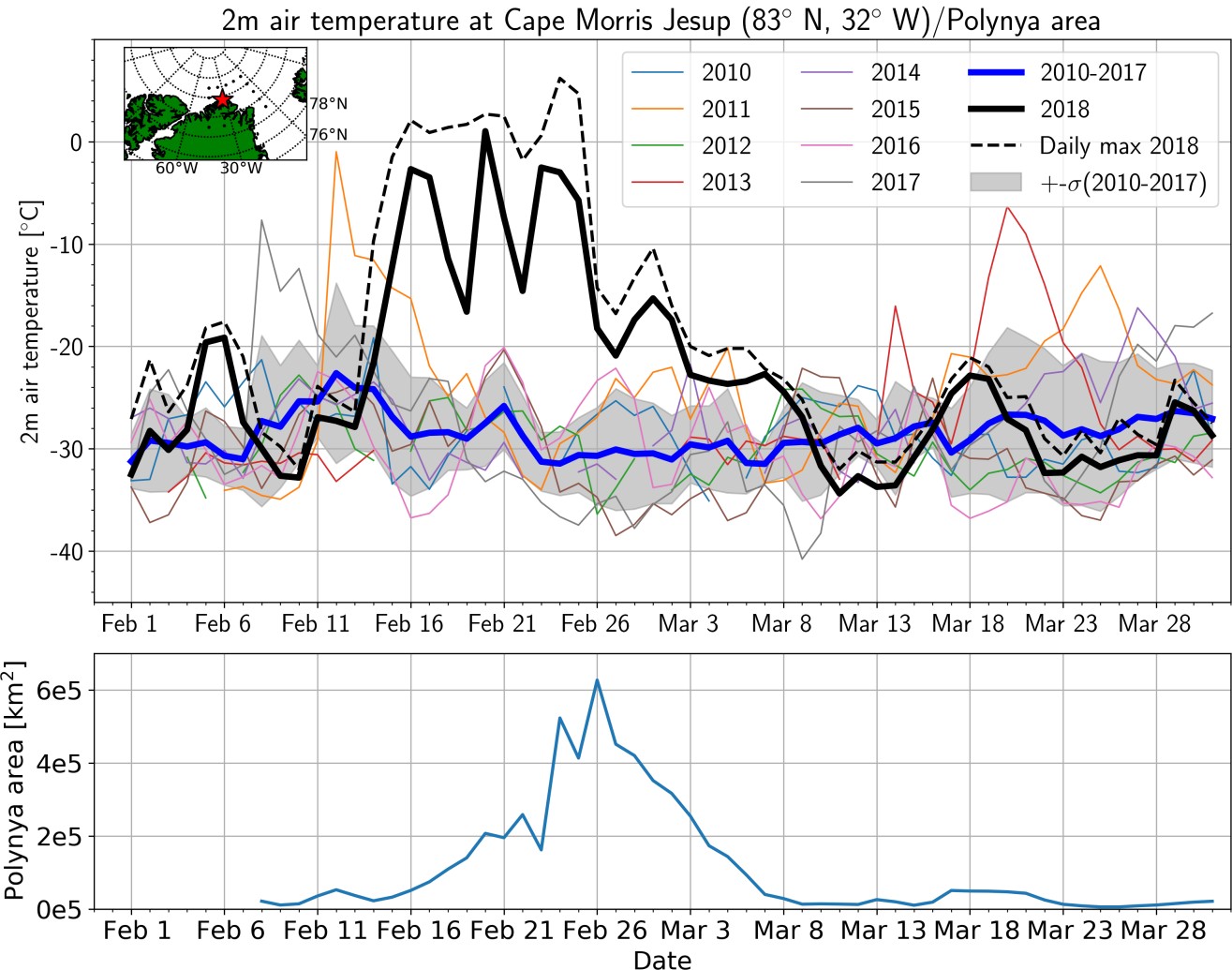

**Figure 6.** Upper plot: Daily average of 2 m air temperature data at the Danish Meteorological Institute's weather station at Cape Morris Jesup since 2010. The blue line shows the mean air temperature of the years 2010–2017, the black line shows the air temperatures in 2018. The thin lines represent the single years. The shades indicate the standard deviation of the air temperatures until 2018. The box and star in the map mark the region of the polynya event and the location of the Cape Morris Jesup Station, respectively. Lower plot: Time series of the polynya area from Fig. 5, but with daily means instead of all granules.

## 5.2 Sea ice drift

OSI-SAF sea ice drift data between the opening of the polynya (February 14th) and the end of our study period (March 31st) are used to investigate the dynamic drivers of the polynya. In general, the Transpolar Drift exports the sea ice to the Atlantic

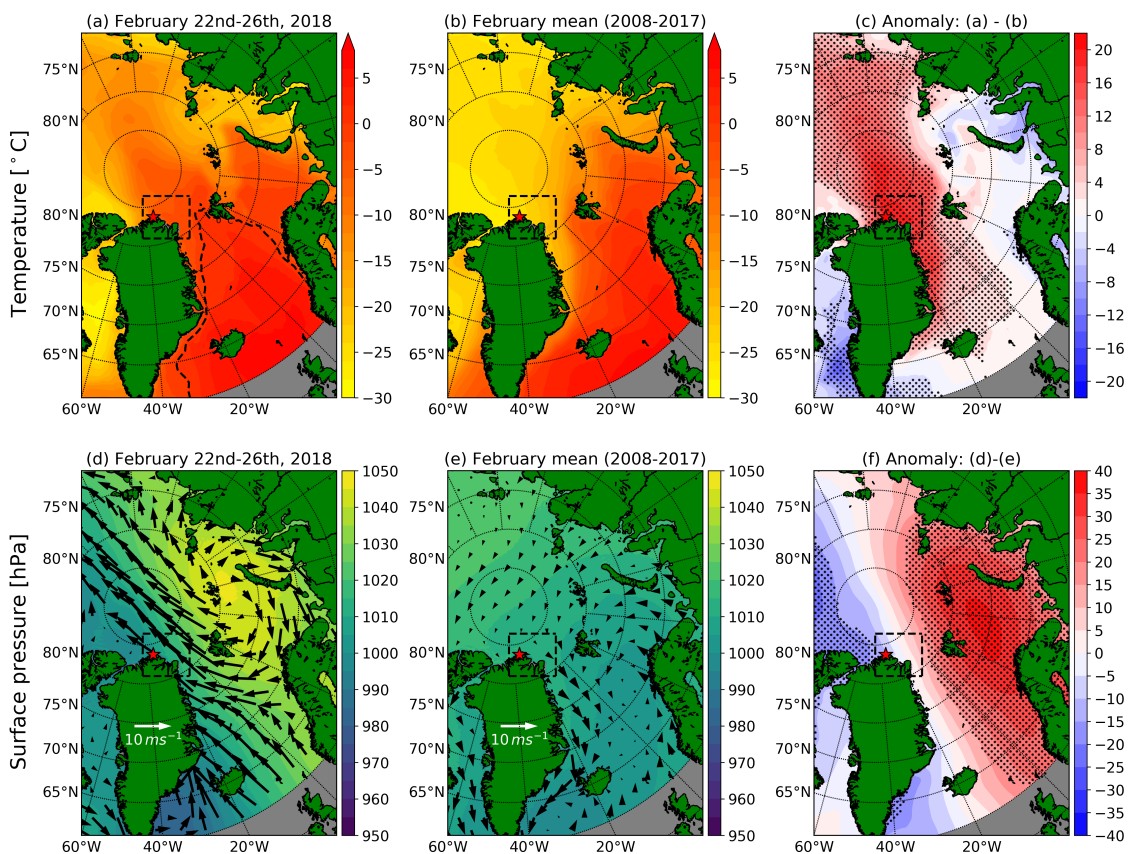

**Figure 7.** (a) ERA5 2 m air temperature from February 22nd to February 26th, 2018, together with the $0°$ C isotherm. The box and star in the map mark the region of the polynya event and the location of the Cape Morris Jesup Station, respectively. (b) February mean ERA5 2 m air temperature between 2008 and 2017. The $0°$ C isotherm is shown as dashed line. (c) Difference between (a) and (b). Black dots mark points where the air temperature in 2018 was more than two standard deviations above/beneath the 2008–2017 average. (d) ERA5 surface air pressure distribution from February 22nd to February 26th, 2018. The black arrows give the ERA5 10 m wind. (e) Same as (d), but for the February mean surface air pressure between 2008 and 2017. (f) Difference between (d) and (e). Black dots mark points where the surface air pressure in 2018 was more than two standard deviations above/beneath the 2008–2017 average.

Ocean via Fram Strait. This year, however, this sea ice drift pattern was reversed. Where there is normally southward flow, there was northward flow while the polynya opened (Fig. 9a,b). The sea ice drift was not only to the opposite direction than

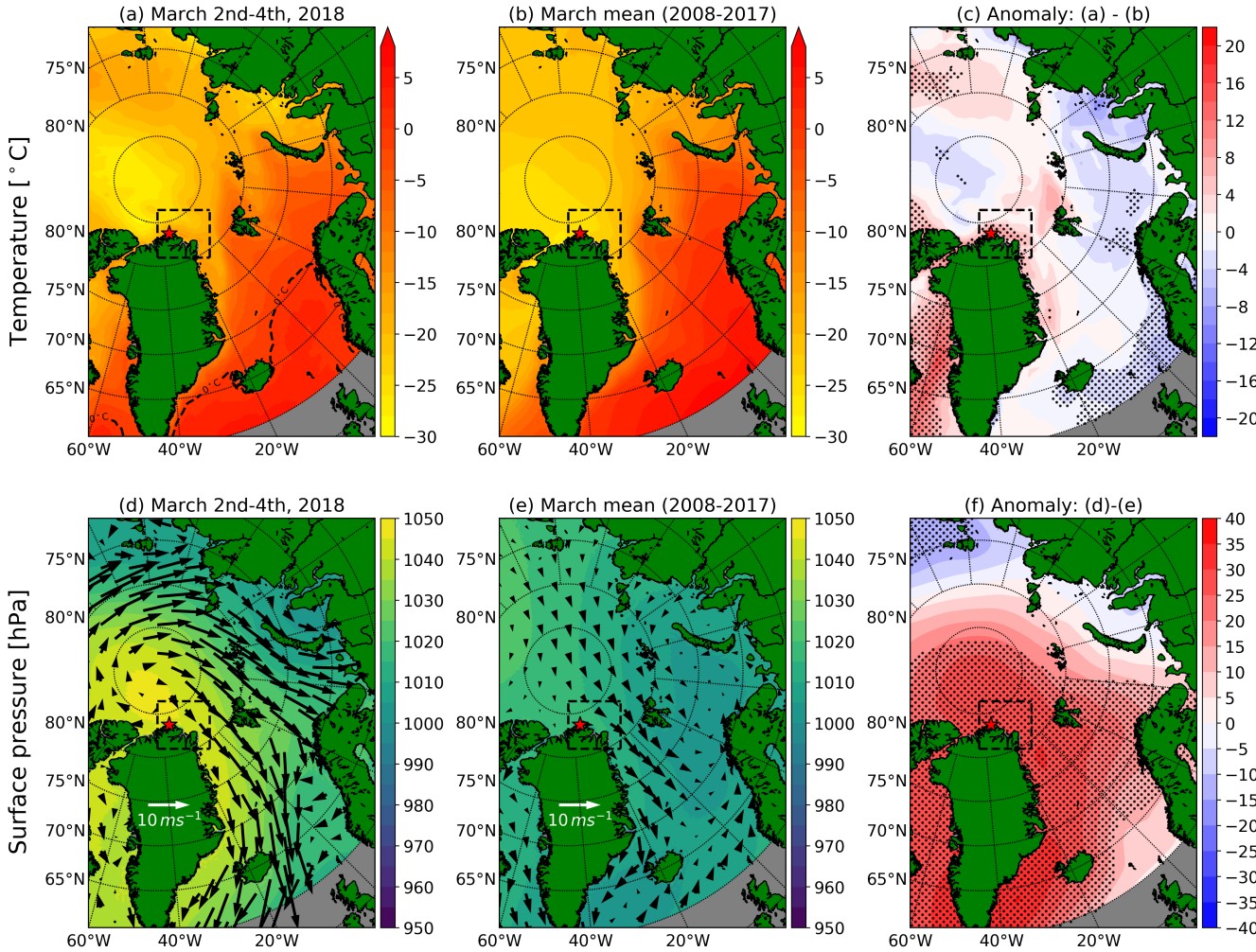

**Figure 8.** (a) and (d): Same as Fig. 7, but for March 2nd–4th. (b) and (e): Same as Fig. 7, but for the March mean. (c) and (f): Difference between (a) and (b) respectively (d) and (e)

usual, but also stronger: The sea ice moved by more than $14\,\mathrm{km}\,\mathrm{d}^{-1}$ over a period of almost two weeks. This is $50\,\%$ more than normal (Table 1). Afterwards, the sea ice drift direction changed to normal conditions, i.e. southeast and there was below-average displacement in the first half of March (Fig. 9c,d). During the second half of March, the sea ice drift was about average (Fig. 9e,f). The mean sea ice displacement and the sea ice drift angle for this year and 2010–2017 are given in Table 1. The sea ice drift angle is the orientation of the sea ice drift towards North. Because of the south-southwestern sea ice drift direction, the sea ice was not completely exported towards the Fram Strait in south-east direction. Instead, it partly returned to the polynya region. Here, it got rafted and ridged with the newly formed sea ice in the polynya. This matches the observation of strong

|  | 2018 | 2010–2017 | 2018 | 2010–2017 |
|---|---|---|---|---|
|  | Displ.[km] | Displ.[km] | Angle[°] | Angle[°] |
| **Feb 14th – Feb 27th** | 14.4 | 8.9±1.9 | 359.4 | 186.5±8.2 |
| **Feb 28th – Mar 16th** | 4.9 | 8.7±1.9 | 157.9 | 187.5±6.2 |
| **Mar 17th – Mar 31st** | 8.7 | 8.7±1.8 | 165.5 | 194.8±12.9 |

**Table 1.** Mean daily displacement and sea ice drift angle in the dashed box shown in Fig. 9(a)–(f), calculated based on the OSI-SAF drift product. For the period 2010–2017, the standard error (defined as the standard deviation divided by the square root of the number of years) is given as well. The angles give the mean orientation of the sea ice drift vectors towards north. The counting goes counterclockwise, so that a sea ice drift angle of $0°/90°/180°/270°$ corresponds to purely north-/west-/south-/eastward movement, respectively.

southwestern sea ice drift between March 16th and March 20th (Fig. 9g). We therefore expect a mix of thermodynamically grown, flat sea ice and rough sea ice grown due to sea ice deformation at the end of March.

The event can be summarised as follows: In February, the sea ice broke apart and was transported northwards. In the first half of March, the sea ice drift was weak and there was rapid thermodynamic sea ice growth in the resulting open water of the polynya since air temperatures were almost $30°$ C below the freezing point (Fig. 6). In the second half of March, parts of the sea ice which had moved northwards in February returned to the area, mainly during one event between March 16th and March 20th, where the sea ice drift was strong and directed towards Southwest, i.e. towards the Northern Greenland coast.

## 6    Processes

This section is dedicated to the processes in the polynya: We estimate the amount of sea ice grown in the polynya and the heat released to the atmosphere. To estimate sea ice growth, we calculate the accumulated thermodynamic sea ice growth assuming calm, snow-free conditions. We employ the freezing degree day parameterisation of Lebedev (1938). The calculations start on February 14th, when the first leads were visible in the merged SIC product. This is compared to airborne electromagnetic (AEM) ice thickness measurements taken on March 30th and 31st and to the simulations of thermodynamic growth by the NAOSIM model. Also, the estimates of thermodynamic growth are compared to the SMOS/SMAP sea ice thickness product of the University of Bremen (Paţilea et al., 2019). For a consistent comparison despite the different grids, we define the polynya area as the area of all pixels which had less than $50\%$ SIC on the respective grid at least once during the event as described in Sect. 2.

The accumulated sea ice growth calculated from the freezing degree days increased strongly over the first days while the polynya opens and then slowed down (Fig. 10). This is expected because the heat flux decreases non-linearly with sea ice thickness once the ice starts to grow. As air temperatures decreased, sea ice growth increased until the accumulated sea ice growth at the end of March was 65 cm. The NAOSIM accumulated sea ice growth increased slowly during the opening of the polynya. Then, it increased strongly from February 25th to March 1st. After that, the accumulated sea ice growth increased slowly to 60 cm at the end of March. During the opening of the polynya, the SMOS/SMAP sea ice thickness is dominated by

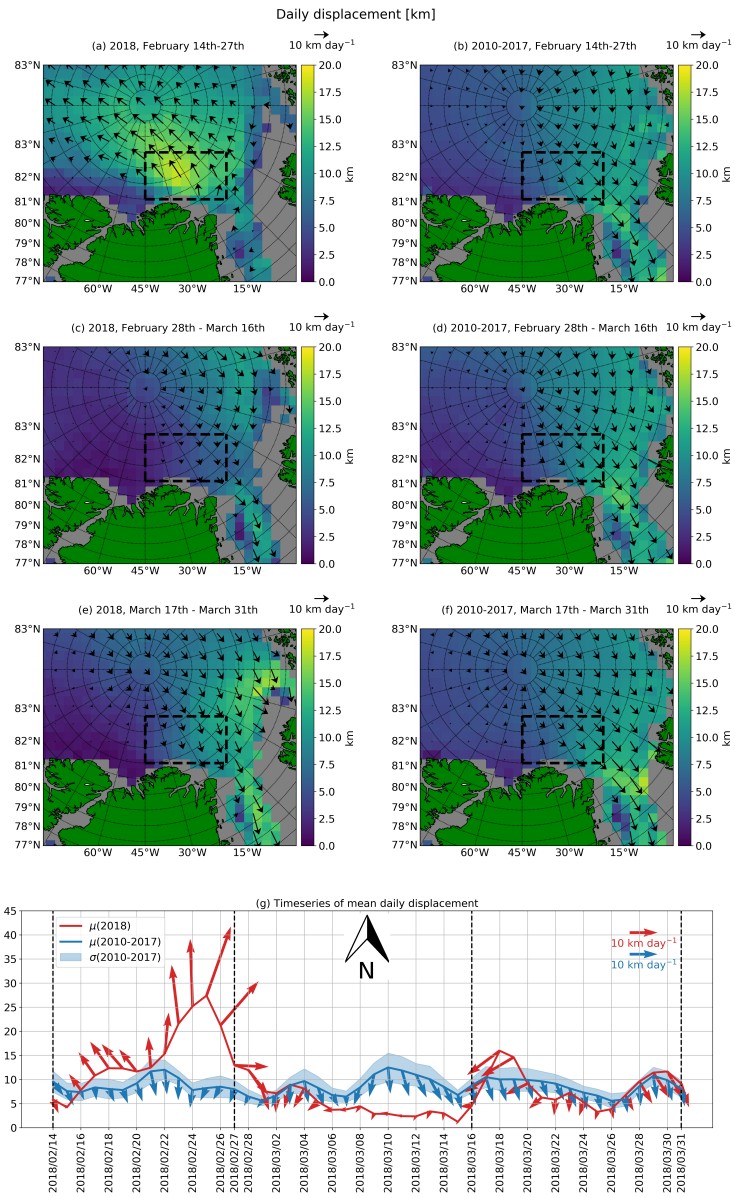

**Figure 9.** Daily displacement based on the OSI-SAF drift product in the polynya region between February 14th and March 31st. The periods February 14th–27th, February 28th–March 16th and March 17th–31st in 2018 are considered in (a), (c) and (e). The same periods, but for the mean between 2010 and 2017 are considered in (b), (d) and (f). The vectors in (a)–(f) show the mean sea ice drift velocity in the corresponding periods, where velocity is defined as displacement per day. (g) shows a time series of the displacement (lines), the standard error of the sea ice drift between 2010 and 2017, defined as standard deviation divided by the square root of the number of years (blue shades) and the sea ice drift velocity (vectors). The orientation is such that an upward-pointing vector points towards North. The arrows in the top right give the scale of the arrow length. Displacement and sea ice drift velocity are the averages in the black dashed boxes in (a)–(f).

dynamic processes. Also, the SMOS/SMAP algorithm assumes 100 % SIC (Paţilea et al., 2019; Huntemann et al., 2014), which is not the case here. In this phase, it can therefore not be compared to the estimates of pure thermodynamic growth. After the refreezing started, it evolved synchronously to the accumulated thermodynamic sea ice growth from the freezing degree day parameterisation until both datasets showed sea ice thicknesses of 50 cm. Since the SMOS/SMAP algorithm does not retrieve

sea ice thicknesses above 50 cm (Huntemann et al., 2014; Paţilea et al., 2019), we can not compare the two thermodynamic estimates to the SMOS/SMAP sea ice thickness product after March 20th (Fig. 10).

  In addition to sea ice growth, we estimate the thermodynamically produced sea ice volume by multiplying the accumulated growth rates from Fig. 10 with the maximum area covered by the polynya. For the maximum area, we again consider all points where the sea ice concentration dropped beneath 50 % at least once during the polynya event. The freezing degree day

parameterisation yields a sea ice volume of 33 km$^3$, NAOSIM yields a sea ice volume of 15 km$^3$. The lower sea ice volume by NAOSIM is because the area of the polynya in the model was only half as big compared to the observations.

  Figure 11 compares the accumulated thermodynamic sea ice growth to the sea ice thickness measured by three AEM flights on March 30th and 31st. Their modal/mean value was 1 m/1.94±1.83 m with a smaller mode at 5 cm. The small mode is caused by refrozen leads covered with dark and light nilas. This explains the presence of classes of very thin ice adjacent to the open

leads. The tail of the frequency distribution in Fig. 11 represents deformed ice rather than purely thermodynamically grown flat ice. We note that there is a difference of the main mode of the AEM measurements of 1.0 m which normally represents the thickness of the most abundant, thermodynamically grown ice (e.g. Haas et al. (2010)), and the 0.60 m and 0.65 m obtained by the NAOSIM and FDD models, respectively. This difference can be due to insufficient heat flux assumptions in the models, in particular unrealistic ocean heat flux, or it can indicate that much of the level ice in the polynya was also formed by rafting,

which could increase level ice thickness much above the thermodynamically achievable thickness. However, the much larger mean AEM sea ice thickness of 0.94 m above the modal sea ice thickness demonstrates the importance of dynamic ice growth by sea ice convergence and compression as a result of the closing of the polynya. The heat released to the atmosphere calculated by the NAOSIM model (Fig. 10b) is closely coupled to the opening and closing of the polynya. It is negative, i.e. directed from the ocean to the atmosphere, throughout the entire event. The average/maximum daily heat flux was -40/-124 W m$^{-2}$. The

time-integrated heat flux was -866 W m$^{-2}$.

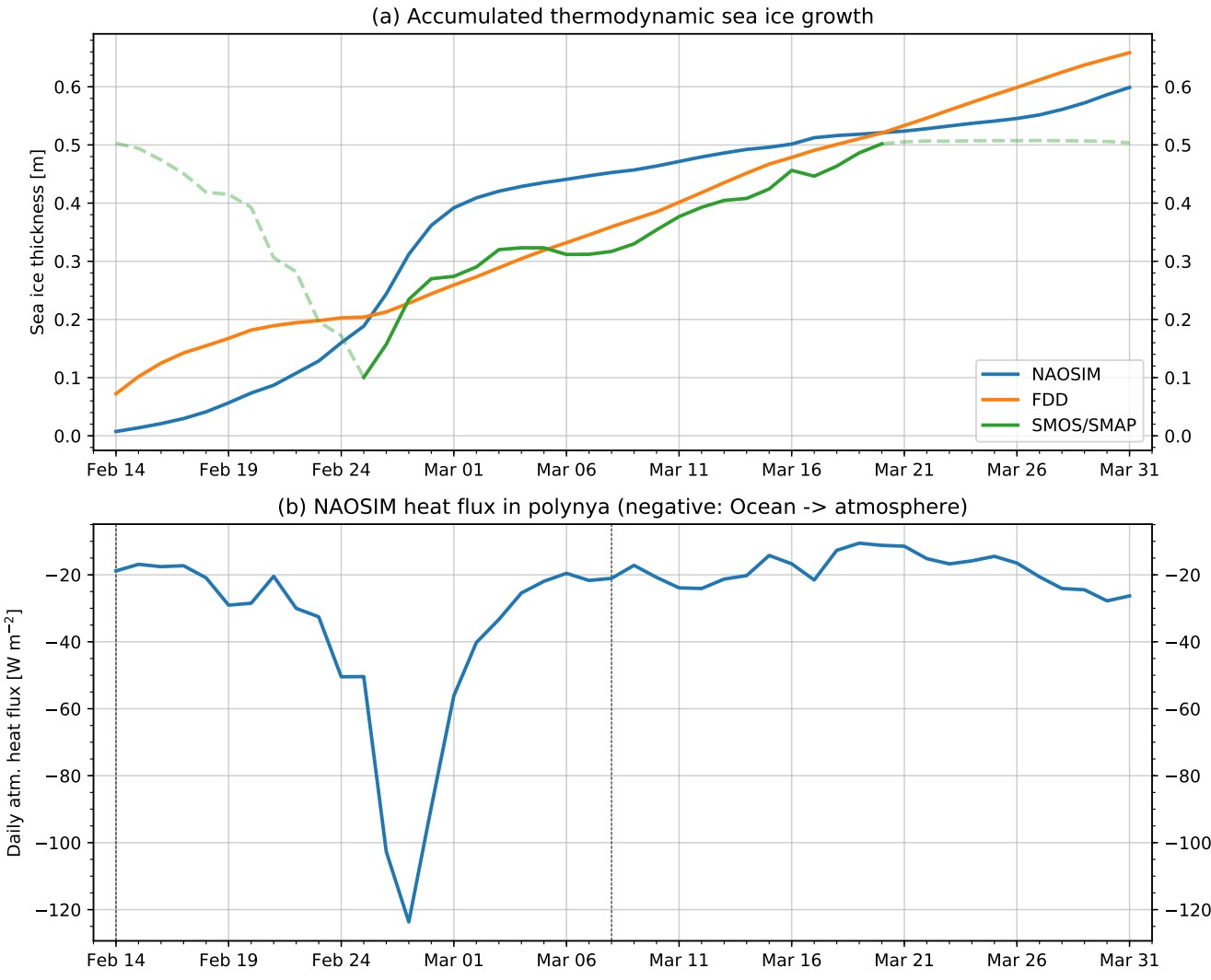

**Figure 10.** (a) Accumulated thermodynamic sea ice growth after February 14th modeled by NAOSIM and estimated using the freezing degree day parameterisation. The SMOS/SMAP passive microwave sea ice thickness product of the University of Bremen is shown for comparison. Days before the refreezing started are marked by the faint dashed line. the same holds for the days after the algorithm reached its maximally retrievable sea ice thickness.(b) Spatially averaged atmospheric heat flux, defined negative upward (blue line) during the polynya period (vertical dashed lines).

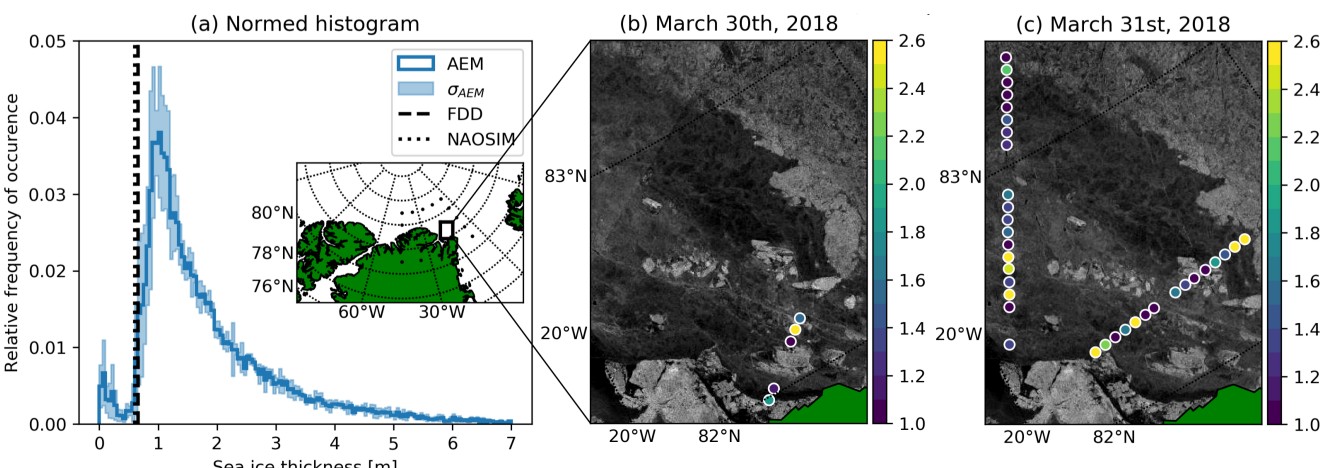

**Figure 11.** AEM measurements of the thickness of young first year ice formed in the polynya until the end of March. (a): Normalised histogram of the AEM measurements. The shades indicate the standard deviation of the three single flights. The black vertical line shows the mean sea ice thickness calculated based on freezing degree days since February 14th. The dotted polygon shows the region of the polynya. The small black rectangle shows the region of the AEM flights. (b) and (c): AEM flights on March 30th and 31st. The dots show the AEM measurements, averaged every 5 km. The background shows a Sentinel-1 mosaic for the respective day.

## 7 Discussion

Comparing the 2018 polynya north of Greenland to the climatology between 1979 and 2017, we find that the SIC in the polynya area during the peak period between February 25th and February 27th were the lowest observed during any day between November and April since 1979. This confirms the findings of Moore et al. (2018) who showed that the mean February SIC 2018 was smaller than any mean February SIC between 1979 and 2018. Our statement is even stronger as we include the entire winter season in our comparison. The 2018 winter polynya thus was a first-time event which had influence on the regional sea ice production and ocean-to-atmosphere heat flux as discussed below. An event as strong as this never happened before during the satellite period. The 1979–2017 all-time minima curve accumulates all potential polynya events during that period. We identify clear but smaller polynya events like in 1986 in mid December. Improved monitoring of these events is possible with the high-resolution, spatially continuous SIC dataset which we present in this paper.

By comparing the merged, MODIS and AMSR2 SIC, we find that AMSR2 SIC is higher than MODIS for high SIC. As the merged SIC preserve the AMSR2 SIC mean, this also holds for the merged SIC. The differences between the MODIS and the AMSR2 SIC arise because the algorithms have different sensitivities to sea ice thickness: AMSR2 is based on the polarisation difference which is independent of the sea ice thickness if the sea ice is thicker than 10 cm (Heygster et al., 2014). MODIS SIC is based on the local sea ice surface temperature anomalies and assumes a bimodal sea ice thickness distribution (Drüe

and Heinemann, 2004). If the sea ice thickness varies while the SIC is 100 %, the SIC of thin sea ice will thus be reported as reduced SIC. Within the 48 by 48 km window used for the derivation of the sea ice tiepoint, the sea ice thickness is expected to vary. This is also the reason why the MODIS SIC are lower than the merged SIC before and after the maximal extension of the polynya (Fig. 3). The SIC underestimation is tolerated by Drüe and Heinemann (2004) because the algorithm was designed to

derive the thermal surface conditions rather than the physical surface conditions and the oceanic heat flux depends on sea ice thickness. However, the SIC underestimation causes a discrepancy when compared to AMSR2 SIC. The described dependence of the MODIS SIC on sea ice thickness and the fact that the SAR SIC also is close to 100 % let us conclude that the AMSR2 SIC are closer to the true SIC here. Thus we merge the AMSR2 SIC and MODIS SIC keeping the AMSR2 SIC on a larger scale but adding the higher resolution of the MODIS SIC to resolve smaller leads and openings.

Comparing the histograms of the SIC datasets, we find that the leads west of the polynya tend to be smeared out by the AMSR2 SIC. This is caused by the lower spatial resolution of AMSR2 and not a deficiency of the algorithm. It illustrates that the merged SIC are better than AMSR2 or MODIS SIC alone. Using only MODIS SIC would mean an underestimation of the SIC in many cases, as described above, and would be limited to cloud-free scenes. Using only AMSR2 SIC would result in smearing out narrow leads. Also, refrozen leads which are covered by snow or sea ice which is thicker than 10 cm would not

be identified. The merged product's magnitude is closer to the SAR SIC than the MODIS SIC and at the same time it preserves most of the high-resolution spatial information of the MODIS data. The SAR SIC themselves are well-suited as a reference product above the region west of the polynya due to their high spatial resolution. However, larger open water areas like the polynya itself can be misclassified due to, e.g., wind roughening effects. Also, SAR data are only available locally. Thus, the merged SIC are the only product which combines high spatial resolution, spatial coverage and daily Arctic-wide coverage.

Over the polynya region, we find that the SAR and MODIS SIC are higher than the AMSR2 SIC. As the air temperatures were still below freezing, it is likely that sea ice production started shortly after the opening. The wind and sea ice drift patterns hindered the evolution of a homogeneous sea ice cover and the newly formed sea ice was turned into grease ice. Under these circumstances, it may be that AMSR2 does not retrieve the grease ice. The grease ice would, however, change the backscatter signature so that the polynya is no longer recognised as such by the SAR SIC. Additionally, a rough water surface can be

misinterpreted as ice by the SAR SIC algorithm as it was trained to retrieve small leads which generally have a smooth surface. The grease ice also shows up as increased MODIS SIC. Another reason for higher MODIS SIC is that the sea ice tiepoint is derived based on the local sea ice surface temperature anomaly. If the surrounding sea ice surface temperature is only slightly below freezing, the range between the dynamic sea ice tiepoint and the fixed water tiepoint gets small and small sea ice surface temperature variations cause high SIC variations. The described sensitivity of the MODIS and SAR SIC towards very freshly

grown sea ice is also the reason why they are lower during the maximal extension of the polynya (Fig. 3). It may be that we underestimate the SIC here by tuning MODIS SIC to the AMSR2 SIC as we get SIC between 0 % and 20 %, although the actual concentration of grease ice is probably higher. We tolerate this as a tradeoff because the approach allows better retrieval of higher SIC and a spatially continuous field. The advantage of the higher resolution of the merged SIC product was shown in Figure 4. It was most pronounced during the early break-up, when the open water extent of the merged SIC was up to 60,% 

higher than that of the AMSR2 SIC. The reason is that small leads which are formed while the polynya breaks up are resolved

by the merged product, but not or only hardly by the AMSR2 SIC. This is caused partly by the higher resolution of the merged product and partly because the MODIS SIC can retrieve refrozen leads which are not retrieved by the AMSR2 SIC any more.

Next, we look into the spatial and temporal evolution of the polynya and the environmental conditions. At its maximal extent, it spanned more than $60,000\,\mathrm{km^2}$. The mean size during the opening was $11,000\,\mathrm{km^2}$. This is slightly larger than the average size of 17 recurring Arctic polynyas reported by Preußer et al. (2016). They find sizes between 400 and $43,600\,\mathrm{km^2}$. The opening of the polynya was driven by anomalous sea ice drift. It was directed northwards where it is normally directed southwards. Besides, the sea ice drift speed was $14\,\mathrm{km\ d^{-1}}$, which is $50\,\%$ stronger than in the eight years before. Also, other studies (Kwok et al., 2013; Vaughan et al., 2013) find typical sea ice drift speeds between five and $10\,\mathrm{km\ d^{-1}}$ in this region. During the second half of the opening period, the sea ice drift anomaly was caused by a persistent high-pressure system above the Eurasian Arctic. The drift pattern during the polynya event has also been analysed by Moore et al. (2018). They use data of the Pan-Arctic Ice Ocean Modeling and Assimilation System (PIOMAS, Zhang and Rothrock (2003)). We can only compare our findings indirectly because they look at the sea ice thickness change due to sea ice motion while we look at the sea ice drift directly. Still, the temporal evolution is consistent: They identify a first peak on February 16th and a second, stronger one on February 23rd and 24th. In our sea ice drift time series, the peaks are one to two days later. The event on March 3rd during which the sea ice was partly returned to the polynya area is also visible in their data. Since their time series ends on March 5th, further comparison is not possible. We found that the surface air pressure distribution during February and March was coincident with the opening and closing of the polynya. Moore et al. (2018) identify the surface pressure distribution and the associated warm-air intrusion as surface response to a sudden stratospheric warming which occurred in early February. The high-pressure system caused northward winds in the polynya area and increased the opening rate. The warm-air intrusion from the mid-latitudes featured air temperatures up to $20°\,\mathrm{C}$ above the average, visible in both reanalysis data and local measurements. The heat released by the polynya contributed to the anomaly. Other studies (Graham et al., 2017; Woods and Caballero, 2016; Moore, 2016; Mewes and Jacobi, 2018) report that such winter warming events have occurred since the 1950s, but did not last as long and were weaker than in recent years. Also, they were not related to polynyas. Even if the $2\,\mathrm{m}$ air temperatures in our case were exceptionally high, they were below/only slightly above the freezing point. The advected air temperature anomaly contributed to the polynya development only indirectly: It slowed the sea ice growth, but did not prevent sea ice growth totally and did not melt the sea ice. This is again consistent with the results of Moore et al. (2018) who show that the thermodynamic sea ice production was always positive, i.e. no sea ice melt occurred. After the air pressure distribution changed, the sea ice drift was directed towards Fram Strait as usual and air temperatures were $20°\,\mathrm{C}$ below the freezing point. The polynya refroze and closed quickly. While it was opened, it contributed to the air temperature anomaly due to heat release.

We identify two periods of enhanced sea ice drift directed towards the Northern Greenland coast in the beginning and in the second half of March (Fig. 9). These closing events have caused deformation of the newly formed sea ice in the polynya. At the end of March, the polynya was covered by a mixture of second and multiyear ice from before the event, deformed newly grown young ice and flat new ice (Fig. 11). Our estimate of thermodynamic sea ice growth ($60\,\mathrm{cm}$ modeled by NAOSIM, $65\,\mathrm{cm}$ estimated by the freezing degree day parameterisation) for March 31st is thus likely an underestimation of the actual

sea ice thickness due to the sea ice thickening by deformation. This is confirmed by comparing these estimates to AEM sea ice thickness measurements at the end of March, which found a modal sea ice thickness value of 1 m as well as a tail towards higher sea ice thicknesses due to deformation. The SMOS/SMAP algorithm assumes 100 % sea ice concentration. This was not always the case during the event. In fact, our SIC curves in Figure 3 show SIC down to 65 %. This contributes to the SMOS/SMAP sea

ice thickness decrease before February 25th. Especially for very thin sea ice, passive microwave retrievals of sea ice thickness and sea ice concentration are ambiguous and it is hard to disentangle the influence of the two quantities on the signal (Ivanova et al., 2015; Heygster et al., 2014). A quantitative estimate of how much the lower sea ice concentration influenced the sea ice thickness retrieval would be beyond the scope of this paper. We note that the SMOS/SMAP sea ice thickness during the opening and early refreezing are less reliable than at a later stage of refreezing. The SMOS/SMAP sea ice thickness is only valid until a

sea ice thickness of 50 cm (Huntemann et al., 2014; Paţilea et al., 2019). Therefore, we can only compare the SMOS/SMAP sea ice thickness to the other products between February 25th, when the refreezing starts and March 20th, when the SMOS/SMAP sea ice thickness reaches 50 cm. The agreement between the freezing degree day parameterisation, the SMOS/SMAP sea ice thickness and the NAOSIM sea ice thickness in this period is good. For SMOS/SMAP and the freezing day parameterisation, this is partly because the SMOS/SMAP algorithm was trained using this parameterisation (Huntemann et al., 2014; Paţilea

et al., 2019). The influence of the warm-air intrusion on the quality of the sea ice thickness retrieval was probably negligible. The air temperature was only above $0°$ C during the opening of the polynya. During the refreezing phase which we analyse here, the air temperature was beneath $0°$ C and therefore did not influence the sea ice thickness retrieval.

By comparing the estimates of the thermodynamically produced sea ice volume, we find a discrepancy between the freezing degree day parameterisation ($33\,\mathrm{km}^3$) and the NAOSIM model ($15\,\mathrm{km}^3$). The discrepancy is because the polynya in the model

is only half as large as in the observations. A similar finding was presented in Moore et al. (2018), who find that the polynya in the PIOMAS model was significantly smaller than in the observations. Since our observations agreed well with the outline of the polynya in the SAR images, we conclude that the $33\,\mathrm{km}^3$ are the better estimate. Preußer et al. (2016) give January-March accumulated sea ice production rates of $52\,\mathrm{km}^3$ on average for 17 Arctic coastal polynyas. According to Tamura and Ohshima (2011), the ten major coastal polynyas in the Arctic produce between 130 and $840\,\mathrm{km}^3$ per year. Total winter-accumulated

sea ice production in Arctic polynyas has been estimated between $1811\,\mathrm{km}^3$ (Preußer et al., 2016) and $2940\,\mathrm{km}^3$ (Tamura and Ohshima, 2011). However, given that normally sea ice production north of Greenland is negligible and that the 2018 polynya was only open for three weeks while the values of Preußer et al. (2016)/ Tamura and Ohshima (2011) are given for three months/an entire year, the event is still remarkable on a regional scale. Finally, we estimate a mean/maximum heat flux of -40/-124 $\mathrm{Wm}^{-2}$ during the time when the polynya was opened. This is small compared to the heat fluxes given by Morales-

Maqueda et al. (2004). They report mean heat fluxes between -38 and -105 $\mathrm{Wm}^{-2}$. We attribute this to the warm-air intrusion. When the polynya was opening, the air temperatures were around $-10°$ C, so that the heat flux was comparably small. When the air temperatures decreased to $-30°$ C, the polynya had already started to refreeze, which dampened the heat flux.

# 8 Summary & Conclusions

This paper uses a new SIC product at 1 km resolution from merged passive microwave (AMSR2) and thermal infrared (MODIS) data. The product comprises the high spatial resolution of the thermal infrared data and the spatial coverage of the passive microwave data. Its benefit is demonstrated by means of the polynya which opened north of Greenland in winter 2018. We show that the merged product detects more leads than the passive microwave data and at the same time allows continuous monitoring of the event.

The polynya opened in the second half of February. The open water area expanded over 12 days, reached its maximal extent of more than $60,000 \, \mathrm{km}^2$ on February 26th and decreased linearly until it closed on March 8th. The closing was due to fast refreezing after the warm-air intrusion abated. Additionally, there was dynamic closing by southward sea ice drift. The merged SIC show closed sea ice cover after March 8th. Nonetheless, the area of freshly grown sea ice is still distinguishable in SAR images on March 31st.

The evolution is driven by the sea ice drift in the polynya region. The sea ice drift was directed northwards instead of the usually dominating southward direction during the polynya opening. Furthermore, it was $50 \, \%$ stronger than usual. The sea ice drift was weak during the first half of March, allowing for undisturbed thermodynamic growth of new sea ice. Two convergent events at the end of February and mid March brought back sea ice which was exported from the polynya area during the formation. Therefore, there is a mixture of flat, thermodynamically grown and rough sea ice grown due to sea ice dynamics at the end of March.

Temperatures during the opening of the polynya were more than 20° C above average. This was caused partly by a high-pressure system above the Kara Sea which brought in warm air from the Atlantic. However, the air temperatures still remained below freezing. They were not high enough to melt the sea ice, but slowed down the refreezing process. Only locally, the daily maximum air temperature exceeded the freezing point several times, but not long and strong enough to cause substantial melting. The polynya also contributed to this air temperature anomaly due to the heat released from the ocean to the atmosphere.

The questions which we raised in the beginning can be answered as follows:

1. Does merging MODIS thermal infrared and AMSR2 passive microwave SIC allow additional insights about the formation of the polynya?

Before the opening of the polynya, leads were visible in the merged, high-resolution SIC product which would have been smeared out by the AMSR2 SIC due to the coarse resolution. Generally, the merged SIC showed more SIC between $60 \, \%$ and $90 \, \%$ than the AMSR2 SIC, indicative of the leads which are identified due to the higher resolution. Over regions with $100 \, \%$ SIC, an underestimation of the merged SIC compared to the AMSR2 SIC may occur, which we tolerate as a tradeoff for the possibility to resolve more leads. During the opening of the polynya, the merged SIC retrieve up to $60 \, \%$ more open water than the AMSR2 SIC alone.

2. Was the polynya opened thermodynamically or dynamically and how unusual were the environmental conditions?

The polynya was opened by an anomalous sea ice drift event in the end of February, which confirms the findings of Moore et al. (2018). The sea ice drift was directed northwards for 12 days where it is normally directed southwards. Also, it was 50 % stronger than usual. A high-pressure system over the Eurasian Arctic kept the polynya open. It was accompanied by local air temperatures more than 20° C above average, caused partly by advection due to the high-pressure system and partly by heat release from the opening polynya. Although the air temperature was exceptionally high, it was not high enough to melt the sea ice. Events like this have occurred before and are expected to occur more frequently in future due to the expected thinning of the sea ice cover.

3. How much sea ice grew in the polynya and how much heat was released to the atmosphere?

Two estimates of thermodynamic sea ice growth show accumulated growth of 60 cm (NAOSIM model) and 65 cm (freezing degree day parameterisation) on March 31st, i.e. much lower than the modal AEM thickness of 1.0 m. This could indicate that thermodynamic ice growth was strongly underestimated by the models, despite the fact that we have observed a thin snow cover which would have reduced ice growth but was not considered by the freezing degree day parameterisation. On the other hand, the larger observed modal ice thickness could also have been due to rafting which would double the resulting ice thickness compared to the level ice thickness of the undeformed ice at the time of deformation. However, the much larger mean AEM ice thickness of 0.94 m more than the modal ice thickness showed that deformation played a strong role for ice growth in addition to thermodynamic growth considered by the models. According to the freezing degree day parameterisation, 33 km$^3$ of sea ice were produced. The modeled heat release from the ocean to the atmosphere was on average 40 Wm$^{-2}$ while the polynya was open.

## 9 Outlook

The merged SIC product is presented here for the first time. The benefit towards single-sensor SIC is demonstrated. However, more validation is needed and planned in future work. This could be done by using for example high-resolution optical data from the European Union Copernicus Sentinel-2 or the US Landsat satellite. Also, other methods for merging SIC should be tested and compared to the method presented here. For example, Dasgupta and Qu (2006) use a wavelet-based approach to merge MODIS and AMSR-E data for vegetation moisture retrieval and Ricker et al. (2017) use an optimal interpolation scheme to merge CryoSat-2 and SMOS (Soil Moisture and Ocean Salinity) sea ice thicknesses. In principle, their approaches should also be applicable for merging SIC. The combination of passive microwave and thermal infrared data is not expected to work in summer as both of them can not distinguish melt ponds from open water and thus underestimate the summer SIC. Data in the visible wavelength range provide the possibility to detect melt ponds separately (Rösel et al., 2012). Including them into the merging procedure could thus improve the product's performance in summer. Higher resolution can also be achieved by including SAR data as suggested by Karvonen (2014).

At the moment, potential atmospheric effects on the AMSR2 sea ice concentration over sea ice are not considered. Work on correcting these effects is currently undertaken (Lu et al., 2018). It is planned to include this in the future development of our product.

We found that events like this have occurred before north of Greenland. Future research could focus on investigating when, where and how often such events occurred and how strong they were. A polynya was observed in the same spot in August 2018. It could be investigated whether the event described here preconditioned the event in August 2018.

The freeze-up period should also be analysed in more detail. We show that the modal sea ice thickness at the end of March can be approximately reproduced by rather simple approaches neglecting dynamic processes. More research could clarify how much flat and how much rough sea ice there was at the end of March and how the remaining discrepancy between our two estimates and the AEM sea ice thickness of $35\,\mathrm{cm}$ can be explained.

*Code and data availability.* Will be made available before final submission

*Author contributions.* VL, LI and GS designed the research. DM contributed the SAR SIC dataset. FK performed the NAOSIM run. CH contributed the AEM data. VL wrote the paper, except for Sect. 2.2.2 (DM), Sect. 3.4 (CH) and Sect. 2.2.7 (FK). All authors critically discussed the contents and agreed to the submission.

*Competing interests.* The authors declare no competing interests.

*Disclaimer.* TEXT

*Acknowledgements.* This study was supported by the Institutional Strategy of the University of Bremen, funded by the German Excellence Initiative, and by the Deutsche Forschungsgemeinschaft (DFG) through the International Research Training Group IRTG 1904 ArcTrain. We thank the editor, Chris Derksen, for editing the paper. We thank the two anonymous reviewers whose comments greatly helped to improve content, structure and clarity of the paper. We sincerely thank Leif Toudal Pedersen for his constructive and useful comments on this manuscript. We further thank him and www.seaice.dk for the provision of the SAR mosaics. We thank Cătălin Paţilea for the fruitful discussion about the SMOS/SMAP sea ice thickness. We thank Stefan Hendricks and Jan Rohde for the contribution in acquisition and processing of the AEM data. MODIS Ice Surface Temperature data were provided by the National Snow and Ice Data Center (NSIDC) at https://nsidc.org/data/MYD29/versions/6. MODIS geolocation and cloud mask data were provided by the Level-1 and Atmosphere Archive & Distribution System (LAADS) Distributed Active Archive Center (DAAC) at https://ladsweb.modaps.eosdis.nasa.gov/archive/allData/61/MYD03/. We thank JAXA (http://www.jaxa.jp/index_e.html) for the provision of AMSR2 data. We gratefully acknowledge the provision of Copernicus Sentinel data [2018] provided by the European Union via the Copernicus Open Access Hub (scihub.copernicus.eu). ERA5 data were obtained from the Copernicus Climate Change Service Information (2018). The OSI-450 (SIC 1979-2015), OSI-401 (SIC 2015-2018) and OSI-405 (sea ice drift) data were provided by EUMETSAT OSI SAF. Station data from Cape Morris Jesup were obtained at

https://rp5.ru/Weather_in_Cape_Morris_Jesup. We thank the Danish Meteorological Institute for operating the weather station and sharing the data.

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
