# Peer review of "The 2018 North Greenland polynya observed by a newly introduced merged optical and passive microwave sea ice concentration dataset"

_The Cryosphere, 2019_

## Referee Comment (RC1) · Anonymous Referee #1 · 8 Mar 2019

The significance of the research presented is the high-resolution SIC dataset generated from merging of data from optical and microwave sensors. In the Introduction you state "This paper for the first time presents a merged product..." The research has potential use for all polynyas not just this unusual North Greenland polynya of 2018. The title clearly states that a new SIC dataset is created. The first paragraph of the Summary & Conclusions should be the only objective of the paper. Of the questions addressed, only question 1 should be answered in this paper, and probably should be expanded to include how much more information about a polynya can be discovered with this method compared to coarser resolution SIC data sets. Discussion and explanation of this polynya has been presented in Moore et al., (2018). Explanation of this

polynya should not be an objective of this paper. The authors could present explanation and affects, e.g. how much sea ice grew, of this polynya in a different paper. I think the authors should focus on the method of creating their SIC dataset in this paper and that it can contribute to gathering more and/or better information about a polynya. A major revision would probably be necessary to narrow the focus. Sections 5 and 6 could probably be omitted or greatly reduced, as only some general statement(s) on the significance of this unusual polynya event would need to be made.

I was most focused on the new method for SIC and its evaluation as the primary objective of the paper. The reasons for the polynya formation and processes that occurred in it Sections 5 and 6 were less interesting and seem to lack relevance to the new method. I agree that you need to demonstrate the method on a polynya, and that this was an interesting one and you gave a reason for studying it but I think it unnecessary to discuss in detail the environment and processes that happened in this polynya.

Specific comments: Check that all the paragraphs all start with the same indent, or no indent, and space between paragraphs is consistent.

Quality of the graphics is good.

Abstract Line 4 change "which combines" to combining Line 7 change "as" to at Line 13 clarify that it is growth of sea ice thickness

Page 2 line 10 – 11. Give the instrument name AMSR2 first, then the band and algorithm. Page 3 line 30 Change "this February" to February 2018; give the dates that the polynya existed. Page 4 line 1 delete word "additionally" Page 4 line 5 change "we use" to used In both 2.1.1 and 2.1.2 add some information on the orbit characteristics of the platforms and a URL to those mission or instrument home pages. 2.1.1 Are both the swath and daily grid data publically available? 2.1.2 Delete "flies". In some way more clearly state that the data records start in 1999 and 2002. Add URL for NSIDC DAAC for this data product. 2.2.1 EASE-Grid or EASE-Grid2? Explicitly state. 2.2.3 delete "seaice.dk" in first sentence, it's irrelevant there. It is given as the url for the University in

last sentence. 2.2.7 please give name of organization at https://seaice.uni-bremen.de along with the URL 2.2.8 delete the paragraph at end of first sentence. The sentence is not a paragraph by itself.
* * *

---

## Referee Comment (RC2) · Anonymous Referee #2 · 9 Mar 2019

Review of "tc-2019-23" Observation of the 2018 North Greenland polynya with a new merged optical and passive microwave sea ice concentration dataset.

This manuscript presents a new sea-ice concentration methodology, merging an established AMSR2 SIC estimate with thermal-infrared (MODIS) SICs. The new SIC product is then demonstrated in a specific case of the "North Greenland Polynya" event in Feb/March 2018. The polynya event is described with the new SIC data, and additional data (atmosphere reanalysis, ice drift product, coupled ocean/ice model), and the processes leading to re-freeze are described and compared to ground truth (Airborne ElectroMagnetic) data. The event is set in perspective with more typical polynya

events.

It is an interesting paper, that should be published, but both the structure and content of the manuscript must be improved before. I noted several comments (both typos and more general comments) below. I am rather optimistic the authors will -through serious revision work- produce an improved paper, since I found the Discussions were much clearer than the rest of the text. All the material is here, but the text needs are rework.

Title : this paper is really too short on the merging methodology and its evaluation to really consider that we would now have a "new merged optical and passive microwave dataset". I suggest you change your title to underline this is an "introduction" or "initial work" on your product.

Abstract: "20C above the average " → "normal"

Introduction "The recent sea ice retreat ..." : how recent? Also, please add a citation for this sentence. "The 89 GHz sensor..." : change "sensor" with "frequency channels". "3 by 5 km" this is the instantaneous field of view, the effective field of view is closer to "5 by 5 km". "Also, they are insensitive towards the sea ice thickness for thicknesses above 10 cm": this depends strongly on which frequency enters in the algorithms (Ivanova et al. 2015). 10cm might be correct for "near-90GHz" algorithms such as ASI.

About the SAR limitations. Once cloud cover is taken into account, there is much more SAR coverage than thermal infrared. So coverage (swath width and duty cycle) is not a good argument. Automated retrieval of sea-ice from SAR is however a challenging topic, and this might be noted here.

Too many citations of Morales-Maqueda et al. (2004) in 3-4 sentences. Rewrite.

The "oldest and thickest" in the entire Arctic. . . I would expect sea-ice north of CAA to be older. Change to "one of the oldest and thickest"?

The box used for Figure 1 might be too large, as it includes processes in the East Greenland current. Consider using the same box as Moore et al. (2018) which is

better suited (alternatively, justify your box in the text and note it include processes in the East Greenland Sea).

Section 2.1.1 : here again occurences of "3 by 5 km" should be annotated. You are mixing iFoV, eFoV, sampling in the swath, and sampling on the projected grid.

OSI SAF data; 1) if you use OSI-450, the correct citation is Lavergne et al. (2019). 2) it is unclear if you use the box in polar stereographic coordinates (as shown in Fig 1), or in lat/lon box. Clarify. As noted above, consider using the same lat/lon box as Moore et al. (2018). 3) you are stitching together different products (OSI-450 and OSI-401) and should document that the time-series (of average SIC) are consistent at their transition. 4) the uncertainty are probably not spatially independent (Lavergne et al. 2019) and your approach to error bars seem simplistic. Error bars could rather show the variability (standard deviation) of the SIC in the box, this would also be valid.

Description of OSI-SAF sea-ice drift ; "the single-sensor sea ice drift vectors are merged by an optimal _interpolation_ scheme". Also you could cite Lavergne et al (2010) JGRO rather than the ATBD (grey litterature).

End of 2.2.5: "The results therefore represent the thickness of 5 weeks old first-year ice"... add " . . . in those specific environmental conditions".

Move 2.2.5 to section 3 "methods" (since you compute it yourself). However you could add a short subsection to introduce ERA5.

Section 2.2.7, please update your reference to Patilea (now published).

2.2.8 NOASIM: "with the help of an genetic algorithm" change to "a genetic"

Please adopt a more descriptive title for section 4.

Page 10, line 20. If you use OSI-450, the correct reference is Lavergne et al. 2019. Line 21: there are only 30 days in April.

One would expect a nearly 100% average SIC north for Greenland in winter. So it is

either an artefact of the OSI SAF data, or of your box that is too large and includes the East Greenland sea region. Discuss. Also here: you stitched together two SIC products (OSI-450 and OSI-401) and should first comment the temporal consistency of the two, before comparing winter 2017/18 to the climatology.

Avoid "this year's event" and rather refer to the 2017/18 winter season explicitely.

Figure 3 : in what respect is your merged MODIS+AMSR2 SIC much better than the OSISAF SIC (Figure 1)? The OSI-SAF curve on Figure 1 also reaches 0.7 mean SIC. Maybe Mean SIC is not the most appropriate metric here, did you try "open water extent" (1 – sea-ice-extent)? And -again- your box seems very large wrt to the polynya (extending to the East Greenland Sea). Bring the OSI SAF curve from Figure 1 onto Figure 3, and discuss.

Page 15, line 11-13: "Note that the area of the opening is still visible as dark/new ice in the Sentinel-1 mosaic. This shows the limitations of AMSR2 data for the observation of polynya events". I do not understand. If a uniform cover of new-ice is seen in the Sentinel-1 mosaic, then it is correct for AMSR2 SIC to show 100% SIC (irrespective of its thickness if > 10cm). If one is interested in SIC, the SAR image might be hard to turn into 100% SIC, so I would see this as a limitation of the SAR technique. Explain further what you mean, or remove.

Figure 4: lower panel: it would be good to align the design (e.g. yticklabels, gridlines, with that of Figure 5).

You could consider merging together section 4 and 5 because section 4 is quite short.

Section 5: "Given the time of year and the location, only anomalous sea ice drift can be the driver. Nevertheless, a warm-air intrusion (Fig. 5 and 6) contributed to maintaining the polynya open." here you are stating the conclusions of your analysis. Move them to the end of this section, after the presentation of the results.

"We conclude that the sea ice must have been broken up by sea ice drift." Since Moore

et al (2018) exists, you should acknowledge them here, e.g. by stating that you confirm (or not) their conclusions.

Figure 5: fix unit of y-axis for lower panel (should be kmˆ2)

Figure 6 and 7 (d) and (e): please regrid the winds to a polar grid before plotting. Here the lat/lon original sampling is evident and disturb the interpretation (e.g. no vectors in the central Arctic Ocean).

Table 1 and Figure 8: consider re-stating the source of sea-ice drift information (OSI SAF) in the legend, like done in Figure 6 and 7 (ERA5).

Section 6:

We need more details on how the NAOSIM, FDD, and SMOS/SMAP products are compared. The sentence "For consistency, only grid cells with a SIC minimum beneath 50 % during the polynya on the respective grid are considered for the calculation. " in the caption to Figure 9 should be re-written (I did not understand what you mean), and this should be in the text of section 6 (not in the caption). The SMOS/SMAP curve is unsettling because it first drops while the two others grow steadily. Yet, you write "The SMOS/SMAP sea ice thickness evolved synchronously to the accumulated thermodynamic sea ice growth" which is maybe correct after the re-freeze has started, but not before. We need more details on how you extracted the SMOS/SMAP curve (average over the same box)? What you probably want is to show SMOS/SMAP only there there is newly forming ice (not where there is still old sea-ice). Also, the SMOS/SMAP algorithm uses the hypothesis of 100% sea-ice cover (this is not the case during the whole event). And L-band radiometry might (or not, discuss) be affected by the warm air intrusion if there is surface melt. Please re-work this section.

Sea-ice volume computations and Fig 9 (b): first, methodology: "by multiplying the accumulated growth rates from Fig. 9 with the area covered by the polynya"… where do you find the area of the polynya? Is it from Figure 4 and 5 (lower panels)? If the

case, then after March 8th the area is 0 (according to the merged product), so we would expect the volume curves to not grow anymore? Looking again at Figure 9 (b) it almost seems it is a scaling from Figure 9 (b). So did you use a fixed polynya area? If a fixed polynya area, the plot does not bring information compared to your sentence "The freezing degree day parameterisation yields a sea ice volume of 33 km3 , NAOSIM yields a sea ice volume of 15 km3 ." Please rework the description of your "new ice volume" computation.

Page 23 line 6: This would be Figure 9(c).

Figure 10: panels to the right with AEM. Interesting, but difficult to know where we are. Suggestion to plot a larger area or add a larger map as inlet to show where the sampled region is. Also, merge the two maps and use a different symbol (or a label) to show the difference of date. The AEM frequency distribution has a first peak around 0.1m. What is this, an artefact? Describe and comment.

Section 7, discussions: "extraordinary" is often (always?) used in a very positive sense, while you mean here this is a first-time event.

Again, Moore et al. (2018) already compared with the 1978-2017 climatology, so your should put your findings in perspective of their study.

Otherwise, I found the discussion much much clearer than the text in Section 6 (e.g. on the volume computation, the meaning of the heat exchange, etc...).

Page 29, line 30: "European Copernicus Sentinel-2" add "Union".

Acknowledgements: since you use both SIC and SIDrift from EUMETSAT OSI SAF, you could add them in this section. Also, you could credit DMI for running (and sharing data from) the weather station.
* * *

---

## Author Comment (AC1) · 28 May 2019

**Letter to Reviewer 2**

Valentin Ludwig (vludwig@uni-bremen.de), Gunnar Spreen, Christian Haas, Larysa Istomina, Frank Kauker and Dmitrii Murashkin

May 28, 2019

Dear Reviewer,

our sincere thanks for taking your time to assess our paper. Your constructive and encouraging feedback is greatly appreciated. Please find our detailed response to your comments below. Your comments are marked by an **R**, our answers are marked by an **A**.

**R**: Title : this paper is really too short on the merging methodology and its evaluation to really consider that we would now have a "new merged optical and passive microwave dataset". I suggest you change your title to underline this is an "introduction" or "initial work" on your product.
**A**: We have changed the title to "The 2018 North Greenland polynya observed by a newly introduced merged optical and passive microwave sea ice concentration dataset".

**R**: Abstract: "20° C above the average " → "normal"
**A**: We have replaced "20° C above the average" by "20° C higher than normal".

**R**: Introduction "The recent sea ice retreat ..." : how recent? Also, please add a citation for this sentence.
**A**: We specified that we refer to the summer sea ice retreat since 2007 and refer to Dai et al. (2019).

**R**: "The 89 GHz sensor..." : change "sensor" with "frequency channels".
**A**: We have followed your recommendation and exchanged the words.

**R**: "3 by 5 km" this is the instantaneous field of view, the effective field of view is closer to "5 by 5 km".
**A**: We added in the text that "3 by 5 km" refers to the instantaneous field of view.

**R**: "Also, they are insensitive towards the sea ice thickness for thicknesses above 10 cm": this depends strongly on which frequency enters in the algorithms (Ivanova et al. 2015). 10cm might be correct for "near-90GHz" algorithms such as ASI.

**A**: We now explicitly refer to 89 GHz measurements and included the reference to Ivanova et al. (2015).

**R**: About the SAR limitations. Once cloud cover is taken into account, there is much more SAR coverage than thermal infrared. So coverage (swath width and duty cycle) is not a good argument. Automated retrieval of sea-ice from SAR is however a challenging topic, and this might be noted here.
**A**: We included that automated SIC retrieval from SAR data is difficult and refer to Karvonen (2014). We agree that there is more SAR coverage than thermal infrared. We have added this in the paper. However, the SAR data do still not always allow daily Arctic-wide coverage, which is a disadvantage compared to passive microwave data and our merged product. If the reviewer agrees, we would thus like to keep the argument concerning the SAR coverage in the text.

**R**: Too many citations of Morales-Maqueda et al. (2004) in 3-4 sentences. Rewrite.
**A**: The paragraph now reads "There are two types of polynyas: sensible and latent heat polynyas. Morales-Maqueda et al. (2004) describe both types of polynyas in detail. We continue with the description of latent heat polynyas since the one we investigate pertains to this type. Latent heat polynyas normally develop close to the coast due to off-shore winds and/or ocean currents which cause divergent sea ice motion. Sea ice is pushed away from the coast and new frazil/grease ice forms. Single latent heat polynyas produce up to $800\,\mathrm{km^3}$ per year of sea ice (Tamura and Ohshima, 2011). Heat fluxes are typically between $300\,\mathrm{W\,m^{-2}}$ and $500\,\mathrm{W\,m^{-2}}$ (Haid and Timmermann, 2013; Martin et al., 2004)."

**R**: The "oldest and thickest" in the entire Arctic... I would expect sea-ice north of CAA to be older. Change to "one of the oldest and thickest"?
**A**: Done.

**R**: The box used for Figure 1 might be too large, as it includes processes in the East Greenland current. Consider using the same box as Moore et al. (2018) which is better suited (alternatively, justify your box in the text and note it include processes in the East Greenland Sea).
**A**: Thanks for this suggestion, we adapted the box of Moore et al. (2018). The magnitude of the time series did not change much.

**R**: Section 2.1.1 : here again occurences of "3 by 5 km" should be annotated. You are mixing iFoV, eFoV, sampling in the swath, and sampling on the projected grid.
**A**: We changed it to "5 by 5 km" and add that this refers to the effective field of view.

**R**: OSI SAF data; 1) if you use OSI-450, the correct citation is Lavergne et al. (2019).
**A**: The correct citation is used now.

**R**: 2) it is unclear if you use the box in polar stereographic coordinates (as shown in Fig 1), or in lat/lon box. Clarify. As noted above, consider using the same lat/lon box as Moore et al. (2018).
**A**: We use geographic coordinates and now state this in the text. Also, the same box as in Moore et al. (2018) is used.

**R**: 3) you are stitching together different products (OSI-450 and OSI-401) and should document that the time-series (of average SIC) are consistent at their transition.
**A**: Done.

**R**: 4) the uncertainty are probably not spatially independent (Lavergne et al. 2019) and your approach to error bars seem simplistic. Error bars could rather show the variability (standard deviation) of the SIC in the box, this would also be valid.

**A**: Thank you for pointing out that the assumption of spatial independence does not hold. We have removed the error bars. We think that, if we include the standard deviation within the box, we would also need to include error bars for the mean standard deviation of the climatology for comparison. This would, however, overload the plot in our opinion. Therefore, we suggest to add a second panel to the Figure. The second panel would show the time series of the standard deviation in the box for 2018 together with the mean standard deviation in the box between 1979 and 2017. Figure 1 would then look like this:

[Figure]

Figure 1: Upper panel: Mean OSI-SAF SIC (Lavergne et al., 2019) in the polynya region (indicated by the dashed box on the map in the lower right corner). The black line shows the mean SIC in 2018. The blue line shows the mean SIC between 1979 and 2017. The dark/light shades indicate the 1-/2$\sigma$ interval, respectively. The red line shows the minimal mean SIC between 1979 and 2017 for each day. Lower panel: Time series of the standard deviation in the polynya region for 2018 (black). The blue line shows the mean of the standard deviations in the polynya region between 1979 and 2017.

**R**: Description of OSI-SAF sea-ice drift ; "the single-sensor sea ice drift vectors are merged by an optimal _interpolation_ scheme". Also you could cite Lavergne et al (2010) JGRO rather than the ATBD (grey litterature).
**A**: Both done.

**R**: End of 2.2.5: "The results therefore represent the thickness of 5 weeks old first-year ice"... add "...in those specific environmental conditions".
**A**: Done.

**R**: Move 2.2.5 to section 3 "methods" (since you compute it yourself).
**A**: Done.

**R**: However you could add a short subsection to introduce ERA5.
**A**: We have added the sentences "The ERA5 reanalysis is run at the European Centre for Medium-Range Weather Forecasts (ECMWF). It is the fifth generation of reanalyses from ECMWF. Hourly reanalysis data of $2\,\mathrm{m}$ air temperature and $10\,\mathrm{m}$ wind are available in near-real time at a spatial resolution of $31\,\mathrm{km}$ (Hersbach and Dee, 2016).". Since we describe the data in the same subsection, we decided to describe ERA5 in the same subsection and not in a separate one.

**R**: Section 2.2.7, please update your reference to Patilea (now published).
**A**: Done.

**R**: 2.2.8 NOASIM: "with the help of an genetic algorithm" change to "a genetic"
**A**: Done.

**R**: Please adopt a more descriptive title for section 4.
**A**: We changed the title to 'Sea Ice Concentration'.

**R**: Page 10, line 20. If you use OSI-450, the correct reference is Lavergne et al. 2019.
**A**: The reference was changed to Lavergne et al. (2019).

**R**: Line 21: there are only 30 days in April.
**A**: Corrected, thanks.

**R**: One would expect a nearly 100% average SIC north for Greenland in winter. So it is either an artefact of the OSI SAF data, or of your box that is too large and includes the East Greenland sea region. Discuss.
**A**: The magnitude of the OSI-SAF SIC did not change much after adapting the box of Moore et al. (2018). We visually inspected the time series of average SIC for each year of the climatology. We found that the mean was seldom close to $100\,\%$. The years before

1987 had a slightly lower mean than the years after 1987. Before 1987, SMMR data were used, while SSM/I data were used after 1987, so that there may be a problem with the SMMR data. However, years with mean SIC close to 90 % also occurred after 1987, so that this can not be the only reason. We conclude that this is a shortcoming of the OSI-SAF SIC, but does not contradict our statement that the SIC of this year was very significantly lower than normal.

**R**:Also here: you stitched together two SIC products (OSI-450 and OSI-401) and should first comment the temporal consistency of the two, before comparing winter 2017/18 to the climatology.
**A**: Done.

**R**: Avoid "this year's event" and rather refer to the 2017/18 winter season explicitly.
**A**: Done.

**R**: Figure 3 : in what respect is your merged MODIS+AMSR2 SIC much better than the OSISAF SIC (Figure 1)? The OSI-SAF curve on Figure 1 also reaches 0.7 mean SIC. Maybe Mean SIC is not the most appropriate metric here, did you try "open water extent" (1 – sea-ice-extent)? And -again- your box seems very large wrt to the polynya (extending to the East Greenland Sea). Bring the OSI SAF curve from Figure 1 onto Figure 3, and discuss.
**A**: We have calculated the open water extent for the merged SIC and the AMSR2 SIC and included in a Figure of its own (Figure 2 in this document). The same box as in Moore et al. (2018) was used. We constrain our comparison to cloud-free pixels and to scenes where at least half of the box was covered. The open water extent is normalised to the number of cloud-free pixels. The time series in Figure 2 shows that the absolute differences are between 1 and 3 %. As the benefit of the merged product's higher resolution lies mainly in the retrieval of the leads which form when the polynya starts to open, we also show the quotient of the merged and the AMSR2 SIC during the opening phase of the polynya (lower panel in Figure 2. Here, we see that the merged product is able to retrieve up to 60 % more open water than the AMSR2 SIC alone. This reflects the leads which the AMSR2 SIC do not retrieve.
It is hard to compare this to the OSI-SAF SIC for two reasons. The first one is the low bias of the OSI-SAF SIC which we mentioned above. This makes it more likely for OSI-SAF to have a high open water extent despite its coarse resolution. The second one is that we would need to compare daily means (OSI-SAF) to single overflights (merged SIC). Since we want to focus on the benefit introduced by the merged product's higher resolution, we decided to not include the OSI-SAF and hope that you agree with us in this point.

[Figure]

Figure 2: Time series of the merged and AMSR2 open water fraction. The upper panel shows the absolute value of the open water fraction for both datasets. The middle panel shows the difference of the merged and the AMSR2 open water fraction. The lower panel shows the quotient of the two datasets during the opening phase. Mind the different time span of the lower panel. Only points where MODIS data were available were considered.

**R**: Page 15, line 11-13: "Note that the area of the opening is still visible as dark/new ice in the Sentinel-1 mosaic. This shows the limitations of AMSR2 data for the observation of polynya events". I do not understand. If a uniform cover of new-ice is seen in the Sentinel-1 mosaic, then it is correct for AMSR2 SIC to show 100% SIC (irrespective of its thickness if > 10cm). If one is interested in SIC, the SAR image might be hard to turn into 100% SIC, so I would see this as a limitation of the SAR technique. Explain further what you mean, or remove.

**A**: What we meant was that the SAR data allow identification of the polynya region even after the polynya is refrozen, while the AMSR2 SIC do not. This is, however, not the focus of our paper, so we decided to remove this sentence.

**R**: Figure 4: lower panel: it would be good to align the design (e.g. yticklabels, gridlines, with that of Figure 5).

**A**: Done. The only remaining differences are that Figure 4 shows a time series of all merged overflights while Figure 5 shows daily means. Also, the acquisition times of the four maps shown in Figure 4 are marked in Figure 4 i). The next Figure shows the new version of Figure 4i (upper panel) and Figure 5 for comparison.

[Figure]

(a) New version of Figure 4i: Time series of the polynya area. The polynya area is calculated as sum of the open water fraction (1 - merged SIC) in the map area, multiplied with the respective grid cell size. All available granules are shown. The acquisition times of the 4 maps shown in the paper are marked by the vertical dashed lines.

[Figure]

(b) Lower panel of Figure 5: Same time series as in upper panel, but with daily means instead of all granules.

Figure 3: New version of polynya area time series: The layout of Figure 4i (upper panel here) has been adapted to match that of Figure 5 (lower panel here).

**R**: You could consider merging together section 4 and 5 because section 4 is quite short.

**A**: If the reviewer agrees, we would like to keep the sections separated. We like the structure of having one section (section 4) which presents and discusses the SIC products and the polynya itself, one section (section 5) which explains how the polynya opened and one section (section 6) which focuses on the sea ice production in the polynya.

**R**: Section 5: "Given the time of year and the location, only anomalous sea ice drift can be the driver. Nevertheless, a warm-air intrusion (Fig. 5 and 6) contributed to maintaining the polynya open." here you are stating the conclusions of your analysis. Move them to the end of this section, after the presentation of the results.
**A**: We have moved them to the end, as suggested.

**R**: "We conclude that the sea ice must have been broken up by sea ice drift." Since Moore

et al (2018) exists, you should acknowledge them here, e.g. by stating that you confirm (or not) their conclusions.

**A**: We have added to Section 7 (Discussion) that our findings are consistent with those by Moore et al. (2018). They find that the sea ice thickness loss due to sea ice motion is much higher than the thermodynamical sea ice growth. Also, their time series of ice loss due to ice motion is consistent with our drift time series.

**R**: Figure 5: fix unit of y-axis for lower panel (should be km^2)

**A**: Fixed.

**R**: Figure 6 and 7 (d) and (e): please regrid the winds to a polar grid before plotting. Here the lat/lon original sampling is evident and disturb the interpretation (e.g. no vectors in the central Arctic Ocean).

**A**: We have regridded the winds.

**R**: Table 1 and Figure 8: consider re-stating the source of sea-ice drift information (OSI SAF) in the legend, like done in Figure 6 and 7 (ERA5).

**A**: Done.

**R**: Section 6: We need more details on how the NAOSIM, FDD, and SMOS/SMAP products are compared. The sentence "For consistency, only grid cells with a SIC minimum beneath 50% during the polynya on the respective grid are considered for the calculation." in the caption to Figure 9 should be re-written (I did not understand what you mean), and this should be in the text of section 6 (not in the caption).

**A**: When comparing the modelled (NAOSIM, FDD) to the observed (SMOS/SMAP) values, we wanted to avoid interpolation from the model grid to the observation grid. We therefore defined the polynya area on each grid as the area of all pixels which had less than 50 % at least once during the polynya period. The SIC from the NAOSIM model was used for the determination of the criterion. For deriving the polynya area of the FDD thermodynamical growth, we used the ERA5 SIC. For deriving the area of the SMOS/SMAP sea ice thickness, we used ASI-AMSR2 values at 12.5 km grid spacing. This information has been added in section 2 (Data).

**R**: The SMOS/SMAP curve is unsettling because it first drops while the two others grow steadily. Yet, you write "The SMOS/SMAP sea ice thickness evolved synchronously to the accumulated thermodynamic sea ice growth" which is maybe correct after the re-freeze has started, but not before. We need more details on how you extracted the SMOS/SMAP curve (average over the same box)? What you probably want is to show SMOS/SMAP only there there is newly forming ice (not where there is still old sea-ice).

**A**: Based on the ASI-AMSR2 SIC at 12.5 km grid spacing, we derive a mask of pixels which had less than 50 % SIC at least once during the polynya event. This is the same procedure

as described in our answer above. Meaningful comparison between the SMOS/SMAP sea ice thickness and the NAOSIM and FDD sea ice thickness is only possible after the re-freezing has started and before the sea ice thickness exceeds 50 cm. We therefore plotted the SMOS/SMAP sea ice thickness in a light shade and dashed during the opening phase of the polynya and after it reached 50 cm.

**R**: Also, the SMOS/SMAP algorithm uses the hypothesis of 100% sea-ice cover (this is not the case during the whole event).

**A**: Especially for very low sea ice thickness, it is not possible to disentangle sea ice concentration and sea ice thickness. This decreases the reliability of the SMOS/SMAP sea ice thickness during the very early freeze-up. It is also partly responsible for the decrease of the SMOS/SMAP sea ice thickness while the polynya breaks up. This has been added to the paper.

**R**: And L-band radiometry might (or not, discuss) be affected by the warm air intrusion if there is surface melt.

**A**: The warm air intrusion happened mainly when the polynya was opened. Even then, surface melt likely only occurred sporadically because the 2 m air temperature was only above the freezing point occasionally (Fig. 5, upper panel). The period which is relevant for the comparison of the thermodynamical growth starts on February 25th. At this time, 2 m air temperatures were already -10/-15 °C (daily maximum/mean). Afterwards, it got even colder. We conclude that surface melt did not influence the SMOS/SMAP sea ice thickness retrieval during the phase which we focus on. A corresponding sentence has been added.

**R**: Please re-work this section.

**A**: The section has been rewritten as described above. We hope that it is better understandable now.

**R**: Sea-ice volume computations and Fig 9 (b): first, methodology: "by multiplying the accumulated growth rates from Fig. 9 with the area covered by the polynya": where do you find the area of the polynya? Is it from Figure 4 and 5 (lower panels)? If the case, then after March 8th the area is 0 (according to the merged product), so we would expect the volume curves to not grow anymore?

**A**: We have used the same area which has been used for the sea ice thickness time series in Figure 9. It was derived as the area of all pixels which have been beneath 50 % SIC at least once during the polynya event. Because this is a fixed area, the volume curves increase until the end of March.

**R**: Looking again at Figure 9 (b) it almost seems it is a scaling from Figure 9 (b). So did you use a fixed polynya area? If a fixed polynya area, the plot does not bring information compared to your sentence "The freezing degree day parameterisation yields a sea ice volume of 33 km3 , NAOSIM yields a sea ice volume of 15 km3 ." Please rework the description of your "new ice volume" computation.

**A**: We did indeed use a fixed polynya area, therefore it is a scaling from Figure 9 (a) (we assume that you meant Figure 9 (a) and not Figure 9 (b)). We agree that this plot is not necessary and remove it.

**R**: Page 23 line 6: This would be Figure 9(c).

**A**: Thanks for pointing this out.

**R**: Figure 10: panels to the right with AEM. Interesting, but difficult to know where we are. Suggestion to plot a larger area or add a larger map as inlet to show where the sampled region is.

**A**: We have reshaped the plot and added an inset to Fig. 10a):

[Figure]

Figure 10: AEM measurements of the ice thickness of young first year ice formed in the polynya until the end of March. (a): Normalised histogram of the AEM measurements. The shades indicate the standard deviation of the three single flights. The vertical lines show the mean sea ice thickness calculated based on the freezing degree days (dashed) and the NAOSIM model (dotted) since February 14th. The dotted polygon shows the region of the polynya. The small black rectangle shows the region of the AEM flights. (b) and (c): AEM flights on March 30th and 31st. The dots show the AEM measurements, averaged every 5 km. The background shows a Sentinel-1 mosaic for the respective day.

**R**: Figure 10: Also, merge the two maps and use a different symbol (or a label) to show the difference of date.

**A**: If the reviewer agrees, we would like to keep both maps. The reasons is that they show S1 mosaics for the respective date and it seems more consistent to us to show the mosaic for the corresponding day for each flight.

**R**: Figure 10: The AEM frequency distribution has a first peak around 0.1m. What is this, an artefact? Describe and comment.

**A**: The peak is not an artefact. It is indicative of a small number of leads and very thin ice that was identified by visual inspection. We added the sentences " The small mode is caused by leads which refreeze rapidly to form dark or light nilas. This explains the presence of classes of very thin ice adjacent to the open leads."

**R**: Section 7, discussions: "extraordinary" is often (always?) used in a very positive sense, while you mean here this is a first-time event.

**A**: "an extraordinary event" has been replaced by "a first-time event".

**R**: Again, Moore et al. (2018) already compared with the 1978-2017 climatology, so your should put your findings in perspective of their study.

**A**: We have added to our paper that we confirm and extend the results of Moore et al. (2018). While they show that the mean February SIC in 2018 was lower than any mean February SIC from 1978-2018, we show that this statement holds for the entire period from October 1st to April 30th.

**R**: Page 29, line 30: "European Copernicus Sentinel-2" add "Union".

**A**: Done.

**R**: Acknowledgements: since you use both SIC and SIDrift from EUMETSAT OSI SAF, you could add them in this section. Also, you could credit DMI for running (and sharing data from) the weather station.

**A**: Done.

We hope that we have addressed your comments in a satisfying manner and thank you again for the time you took to assess our manuscript.

Yours sincerely,

Valentin Ludwig (on behalf of the authors)

**References**

Dai, A., Luo, D., Song, M., and Liu, J.: Arctic amplification is caused by sea-ice loss under increasing CO 2, Nature communications, 10, 121, 2019.

Haid, V. and Timmermann, R.: Simulated heat flux and sea ice production at coastal polynyas in the southwestern Weddell Sea, Journal of Geophysical Research: Oceans, 118, 2640–2652, https://doi.org/10.1002/jgrc.20133, URL `https://agupubs.onlinelibrary.wiley.com/doi/abs/10.1002/jgrc.20133`, 2013.

Hersbach, H. and Dee, D.: ERA-5 reanalysis is in production, ECMWF newsletter, p. 7, 2016.

Ivanova, N., Pedersen, L. T., Tonboe, R. T., Kern, S., Heygster, G., Lavergne, T., Sørensen, A., Saldo, R., Dybkjær, G., Brucker, L., and Shokr, M.: Inter-comparison and evaluation of sea ice algorithms: towards further identification of challenges and optimal approach using passive microwave observations, The Cryosphere, 9, 1797–1817, https://doi.org/10.5194/tc-9-1797-2015, URL http://www.the-cryosphere.net/9/1797/2015/, 2015.

Karvonen, J.: A sea ice concentration estimation algorithm utilizing radiometer and SAR data, The Cryosphere, 8, 1639–1650, 2014.

Lavergne, T., Sørensen, A. M., Kern, S., Tonboe, R., Notz, D., Aaboe, S., Bell, L., Dybkjær, G., Eastwood, S., Gabarro, C., Heygster, G., Killie, M. A., Brandt Kreiner, M., Lavelle, J., Saldo, R., Sandven, S., and Pedersen, L. T.: Version 2 of the EUMETSAT OSI SAF and ESA CCI sea-ice concentration climate data records, The Cryosphere, 13, 49–78, https://doi.org/10.5194/tc-13-49-2019, URL https://www.the-cryosphere.net/13/49/2019/, 2019.

Martin, S., Robert, D., Ronald, K., and Benjamin, H.: Estimation of the thin ice thickness and heat flux for the Chukchi Sea Alaskan coast polynya from Special Sensor Microwave/Imager data, 1990–2001, Journal of Geophysical Research: Oceans, 109, https://doi.org/10.1029/2004JC002428, URL https://agupubs.onlinelibrary.wiley.com/doi/abs/10.1029/2004JC002428, 2004.

Moore, G. W. K., Schweiger, A., Zhang, J., and Steele, M.: What Caused the Remarkable February 2018 North Greenland Polynya?, Geophysical Research Letters, 0, https://doi.org/10.1029/2018GL080902, URL https://agupubs.onlinelibrary.wiley.com/doi/abs/10.1029/2018GL080902, 2018.

Morales-Maqueda, M. A., Willmott, A. J., and Biggs, N. R. T.: Polynya Dynamics: a Review of Observations and Modeling, Reviews of Geophysics, 42, https://doi.org/10.1029/2002RG000116, URL https://agupubs.onlinelibrary.wiley.com/doi/abs/10.1029/2002RG000116, 2004.

Tamura, T. and Ohshima, K. I.: Mapping of sea ice production in the Arctic coastal polynyas, Journal of Geophysical Research: Oceans, 116, https://doi.org/10.1029/2010JC006586, URL https://agupubs.onlinelibrary.wiley.com/doi/abs/10.1029/2010JC006586, 2011.

---

## Author Comment (AC2) · 28 May 2019

**Letter to Reviewer 1**

Valentin Ludwig (vludwig@uni-bremen.de), Gunnar Spreen, Christian Haas, Larysa Istomina, Frank Kauker and Dmitrii Murashkin

May 28, 2019

Dear Reviewer,

thanks a lot for your constructive feedback. Your comments helped to improve structure and clarity of our manuscript. Please find our detailed response to your comments below. Your comments are marked by an **R**, our answers are marked by an **A**.

**R**: The research has potential use for all polynyas not just this unusual North Greenland polynya of 2018. The first paragraph of the Summary & Conclusions should be the only objective of the paper. Of the questions addressed, only question 1 should be answered in this paper, and probably should be expanded to include how much more information about a polynya can be discovered with this method compared to coarser resolution SIC data sets. Discussion and explanation of this polynya has been presented in Moore et al., (2018). Explanation of this polynya should not be an objective of this paper. The authors could present explanation and affects, e.g. how much sea ice grew, of this polynya in a different paper. I think the authors should focus on the method of creating their SIC dataset in this paper and that it can contribute to gathering more and/or better information about a polynya. A major revision would probably be necessary to narrow the focus. Sections 5 and 6 could probably be omitted or greatly reduced, as only some general statement(s) on the significance of this unusual polynya event would need to be made. I was most focused on the new method for SIC and its evaluation as the primary objective of the paper. The reasons for the polynya formation and processes that occurred in it Sections 5 and 6 were less interesting and seem to lack relevance to the new method. I agree that you need to demonstrate the method on a polynya, and that this was an interesting one and you gave a reason for studying it but I think it unnecessary to discuss in detail the environment and processes that happened in this polynya.

**A**: We thank you for the suggestion to focus on the merged SIC in this paper. However, we would like to keep the reasons for the polynya and the consequences which it had in the paper, as suggested by Reviewer 2. We are already planning a second paper which focuses on the development and evaluation of our merged SIC. The present paper is intended to be an introduction of the merged SIC together with a case study to show what it can be used for. We like the idea of investigating the polynya event entirely. We also think that we

investigated enough aspects which have not been covered by Moore et al. (2018) to justify including the environmental conditions during the polynya and the consequences which it had in our paper. If you agree, we would like to follow Reviewer 2 and keep sections 5 and 6.

**R**: Check that all the paragraphs all start with the same indent, or no indent, and space between paragraphs is consistent.

**A**: Done.

**R**: Abstract Line 4 change "which combines" to combining Line 7 change "as" to at Line 13 clarify that it is growth of sea ice thickness

**A**: All done.

**R**: Page 2 line 10 – 11. Give the instrument name AMSR2 first, then the band and algorithm.

**A**: Done.

**R**: Page 3 line 30 Change "this February" to February 2018; give the dates that the polynya existed.

**A**: Done.

**R**: Page 4 line 1 delete word "additionally"

**A**: Done.

**R**: Page 4 line 5 change "we use" to used

**A**: Done.

**R**: In both 2.1.1 and 2.1.2 add some information on the orbit characteristics of the platforms and a URL to those mission or instrument home pages.

**A**: We have included the inclination and equator crossing time for the platforms. In 2.1.2, we also added that the Aqua satellite has only 4 minutes time lag compared to GCOM-W1. URLs to the MODIS and AMSR2 home pages have been added.

**R**: 2.1.1 Are both the swath and daily grid data publically available?

**A**: The swath data were processed internally. We have added this in the text.

**R**: 2.1.2 Delete "flies".

**A**: Done.

**R**: In some way more clearly state that the data records start in 1999 and 2002. Add URL for NSIDC DAAC for this data product.

**A**: We now explicitly state that Terra started in 1999 and Aqua in 2002. The URL

`https://nsidc.org/data/MOD29/versions/6` has been added.

**R**: 2.2.1 EASE-Grid or EASE-Grid2? Explicitly state.
**A**: EASE-Grid 2, now explicitly stated.

**R**: 2.2.3 delete "seaice.dk" in first sentence, it's irrelevant there. It is given as the url for the University in last sentence.
**A**: We deleted "seaice.dk".

**R**: 2.2.7 please give name of organisation at https://seaice.uni-bremen.de along with the URL
**A**: We added that it is a product of the University of Bremen.

**R**: 2.2.8 delete the paragraph at end of first sentence. The sentence is not a paragraph by itself.
**A**: Done.

We hope that we have addressed your comments in a satisfying manner. Please do not hesitate to contact us if questions remain.

Yours sincerely,
Valentin Ludwig (on behalf of the authors)

**References**

Moore, G. W. K., Schweiger, A., Zhang, J., and Steele, M.: What Caused the Remarkable February 2018 North Greenland Polynya?, Geophysical Research Letters, 0, https://doi.org/10.1029/2018GL080902, URL `https://agupubs.onlinelibrary.wiley.com/doi/abs/10.1029/2018GL080902`, 2018.

---

## Author Response (AR1)

**Answers to Anonymous Referee 1**

We sincerely thank you for taking your time to review our paper. Your constructive feedback is very much valued. Your comments helped to improve structure and clarity of our manuscript. Please find our detailed response to your comments below. Your comments are marked by an **R**, our answers are marked by an **A**. If applicable, we give the page and line number corresponding to the change in the new manuscript in brackets at the end of our answer in italics. If applicable, we give the page and line number corresponding to the change in the marked-up manuscript version in brackets in italics.

**R**: The research has potential use for all polynyas not just this unusual North Greenland polynya of 2018. The first paragraph of the Summary & Conclusions should be the only objective of the paper. Of the questions addressed, only question 1 should be answered in this paper, and probably should be expanded to include how much more information about a polynya can be discovered with this method compared to coarser resolution SIC data sets. Discussion and explanation of this polynya has been presented in Moore et al., (2018). Explanation of this polynya should not be an objective of this paper. The authors could present explanation and affects, e.g. how much sea ice grew, of this polynya in a different paper. I think the authors should focus on the method of creating their SIC dataset in this paper and that it can contribute to gathering more and/or better information about a polynya. A major revision would probably be necessary to narrow the focus. Sections 5 and 6 could probably be omitted or greatly reduced, as only some general statement(s) on the significance of this unusual polynya event would need to be made. I was most focused on the new method for SIC and its evaluation as the primary objective of the paper. The reasons for the polynya formation and processes that occurred in it Sections 5 and 6 were less interesting and seem to lack relevance to the new method. I agree that you need to demonstrate the method on a polynya, and that this was an interesting one and you gave a reason for studying it but I think it unnecessary to discuss in detail the environment and processes that happened in this polynya.

**A**: We thank you for the suggestion to focus on the merged SIC in this paper. However, we would like to keep the reasons for the polynya and the consequences which it had in the paper, as suggested by Reviewer 2. We are already planning a second paper which focuses on the development and evaluation of our merged SIC. The present paper is intended to be an introduction of the merged SIC together with a case study to show what it can be used for. We like the idea of investigating the polynya event entirely. We also think that we investigated enough aspects which have not been covered by Moore et al. (2018) to justify including the environmental conditions during the polynya and the consequences which it had in our paper. If you agree, we would like to follow Reviewer 2 and keep sections 5 and 6.

**R**: Check that all the paragraphs all start with the same indent, or no indent, and space between paragraphs is consistent.

**A**: Done.

**R**: Abstract Line 4 change "which combines" to combining Line 7 change "as" to at Line 13 clarify that it is growth of sea ice thickness
**A**: All done *(P1 L4, P1 L8, P1 L14)*.

**R**: Page 2 line 10 – 11. Give the instrument name AMSR2 first, then the band and algorithm.
**A**: Done *(P2 L12ff.)*.

**R**: Page 3 line 30 Change "this February" to February 2018; give the dates that the polynya existed.
**A**: Done *(P5 L2f.)*.

**R**: Page 4 line 1 delete word "additionally"
**A**: Done *(P5 L6)*.

**R**: Page 4 line 5 change "we use" to used
**A**: Done *(P5 L1, also P5 L20)*.

**R**: In both 2.1.1 and 2.1.2 add some information on the orbit characteristics of the platforms and a URL to those mission or instrument home pages.
**A**: We have included the inclination and equator crossing time for GCOM-W1 *(P5 L26f.)* and Aqua *(P6 L6ff.)*. In 2.1.2, we also added that the Aqua satellite has only 4 minutes time lag compared to GCOM-W1 *(P6 L6ff.)*. URLs to the AMSR2 and MODIS home pages have been added *(P5 L25f.; P6 L4f.)*.

**R**: 2.1.1 Are both the swath and daily grid data publically available?
**A**: The swath data were processed internally. We have added this in the text *(P6 L2)*.

**R**: 2.1.2 Delete "flies".
**A**: Done *(P6 L4)*.

**R**: In some way more clearly state that the data records start in 1999 and 2002. Add URL for NSIDC DAAC for this data product.
**A**: We now explicitly state that Terra started in 1999 and Aqua in 2002 *(P6 L5)*. The URL `https://nsidc.org/data/MOD29/versions/6` has been added *(P6 L10)*.

**R**: 2.2.1 EASE-Grid or EASE-Grid2? Explicitly state.
**A**: EASE-Grid 2, now explicitly stated *(P6 L19f.)*.

**R**: 2.2.3 delete "seaice.dk" in first sentence, it's irrelevant there. It is given as the url for the University in last sentence.
**A**: We deleted "seaice.dk" *(P7 L9)*.

**R**: 2.2.7 please give name of organisation at https://seaice.uni-bremen.de along with the URL
**A**: We added that it is a product of the University of Bremen *(P8 L24)*.

**R**: 2.2.8 delete the paragraph at end of first sentence. The sentence is not a paragraph by itself.
**A**: The paragraph now also contains a description of how the polynya region has been determined *(P8 L30ff.)*.

We hope that we have addressed your comments in a satisfying manner and thank you again for your assessment of our paper.

**References**

Moore, G. W. K., Schweiger, A., Zhang, J., and Steele, M.: What Caused the Remarkable February 2018 North Greenland Polynya?, Geophysical Research Letters, 0, https://doi.org/10.1029/2018GL080902, URL `https://agupubs.onlinelibrary.wiley.com/doi/abs/10.1029/2018GL080902`, 2018.

**Answers to Anonymous Referee 2**

Please have our sincere thanks for taking your time to assess our paper. Your constructive and encouraging feedback is greatly appreciated. Please find our detailed response to your comments below. Your comments are marked by an **R**, our answers are marked by an **A**. If applicable, we give the page and line number corresponding to the change in the marked-up manuscript version in brackets in italics.

**R**: Title: this paper is really too short on the merging methodology and its evaluation to really consider that we would now have a "new merged optical and passive microwave dataset". I suggest you change your title to underline this is an "introduction" or "initial work" on your product.

**A**: We have changed the title to "The 2018 North Greenland polynya observed by a newly introduced merged optical and passive microwave sea ice concentration dataset".

**R**: Abstract: "20° C above the average " → "normal"

**A**: We have replaced "20° C above the average" by "20° C higher than normal" *(P1 L13)*.

**R**: Introduction "The recent sea ice retreat ..." : how recent? Also, please add a citation for this sentence.

**A**: We specified that we refer to the summer sea ice retreat since 2007 and refer to Dai et al. (2019) *(P2 L4ff.)*.

**R**: "The 89 GHz sensor..." : change "sensor" with "frequency channels".

**A**: We have followed your recommendation and exchanged the words *(P2 L12f.)*.

**R**: "3 by 5 km" this is the instantaneous field of view, the effective field of view is closer to "5 by 5 km".

**A**: We added in the text that "3 by 5 km" refers to the instantaneous field of view *(P2 L15)*.

**R**: "Also, they are insensitive towards the sea ice thickness for thicknesses above 10 cm": this depends strongly on which frequency enters in the algorithms (Ivanova et al. 2015). 10cm might be correct for "near-90GHz" algorithms such as ASI.

**A**: We now explicitly refer to 89 GHz measurements and included the reference to Ivanova et al. (2015) *(P2 L34f.)*.

**R**: About the SAR limitations. Once cloud cover is taken into account, there is much more SAR coverage than thermal infrared. So coverage (swath width and duty cycle) is not a good argument. Automated retrieval of sea-ice from SAR is however a challenging topic, and this might be noted here.

**A**: We included that automated SIC retrieval from SAR data is difficult and refer to Karvonen (2014). We agree that there is more SAR coverage than thermal infrared. We have added this in the paper. However, the SAR data do still not always allow daily Arctic-wide coverage, which is a disadvantage compared to passive microwave data and our merged product. If the reviewer agrees, we would thus like to keep the argument concerning the SAR coverage in the text *(P3 L7ff.)*.

**R**: Too many citations of Morales-Maqueda et al. (2004) in 3-4 sentences. Rewrite.
**A**: The paragraph now reads "There are two types of polynyas: sensible and latent heat polynyas. Morales-Maqueda et al. (2004) describe both types of polynyas in detail. We continue with the description of latent heat polynyas since the one we investigate pertains to this type. Latent heat polynyas normally develop close to the coast due to off-shore winds and/or ocean currents which cause divergent sea ice motion. Sea ice is pushed away from the coast and new frazil/grease ice forms. Single latent heat polynyas produce up to $800\,\text{km}^3$ per year of sea ice (Tamura and Ohshima, 2011). Heat fluxes are typically between $300\,\text{W m}^{-2}$ and $500\,\text{W m}^{-2}$ (Haid and Timmermann, 2013; Martin et al., 2004)." *(P3 L24ff.)*.

**R**: The "oldest and thickest" in the entire Arctic... I would expect sea-ice north of CAA to be older. Change to "one of the oldest and thickest"?
**A**: Done *(P5 L1)*.

**R**: The box used for Figure 1 might be too large, as it includes processes in the East Greenland current. Consider using the same box as Moore et al. (2018) which is better suited (alternatively, justify your box in the text and note it include processes in the East Greenland Sea).
**A**: Thanks for this suggestion, we adapted the box of Moore et al. (2018). The magnitude of the time series did not change much.

**R**: Section 2.1.1 : here again occurences of "3 by 5 km" should be annotated. You are mixing iFoV, eFoV, sampling in the swath, and sampling on the projected grid.
**A**: We changed it to "5 by 5 km" and add that this refers to the effective field of view *(P5 L28f.)*.

**R**: OSI SAF data; 1) if you use OSI-450, the correct citation is Lavergne et al. (2019).
**A**: The correct citation is used now *(P6 L18)*.

**R**: 2) it is unclear if you use the box in polar stereographic coordinates (as shown in Fig 1), or in lat/lon box. Clarify. As noted above, consider using the same lat/lon box as Moore et al. (2018).
**A**: We use geographic coordinates and now state this in the text *(P6 L24)*. Also, the same box as in Moore et al. (2018) is used.

**R**: 3) you are stitching together different products (OSI-450 and OSI-401) and should document that the time-series (of average SIC) are consistent at their transition.

**A**: Done *(P6 L20f.)*.

**R**: 4) the uncertainty are probably not spatially independent (Lavergne et al. 2019) and your approach to error bars seem simplistic. Error bars could rather show the variability (standard deviation) of the SIC in the box, this would also be valid.

**A**: Thank you for pointing out that the assumption of spatial independence does not hold. We have removed the error bars. We think that, if we include the standard deviation within the box, we would also need to include error bars for the mean standard deviation of the climatology for comparison. This would, however, overload the plot in our opinion. Therefore, we suggest to add a second panel to the Figure. The second panel would show the time series of the standard deviation in the box for 2018 together with the mean standard deviation in the box between 1979 and 2017. Figure 1 would then look like this:

[Figure]

Figure 1: Upper panel: Mean OSI-SAF SIC (Lavergne et al., 2019) in the polynya region (indicated by the dashed box on the map in the lower right corner). The black line shows the mean SIC in 2018. The blue line shows the mean SIC between 1979 and 2017. The dark/light shades indicate the 1-/2$\sigma$ interval, respectively. The red line shows the minimal mean SIC between 1979 and 2017 for each day. Lower panel: Time series of the standard deviation in the polynya region for 2018 (black). The blue line shows the mean of the standard deviations in the polynya region between 1979 and 2017.

**R**: Description of OSI-SAF sea-ice drift ; "the single-sensor sea ice drift vectors are merged by an optimal _interpolation_ scheme". Also you could cite Lavergne et al (2010) JGRO rather than the ATBD (grey litterature).
**A**: Both done *(P7 L20; P7 L16)*.

**R**: End of 2.2.5: "The results therefore represent the thickness of 5 weeks old first-year ice"... add "...in those specific environmental conditions".
**A**: Done *(P12 L3)*.

**R**: Move 2.2.5 to section 3 "methods" (since you compute it yourself).
**A**: Done *(P11 L23ff.)*.

**R**: However you could add a short subsection to introduce ERA5.
**A**: We have added the sentences "The ERA5 reanalysis is run at the European Centre for Medium-Range Weather Forecasts (ECMWF). It is the fifth generation of reanalyses from ECMWF. Hourly reanalysis data of $2\,m$ air temperature and $10\,m$ wind are available in near-real time at a spatial resolution of $31\,km$ (Hersbach and Dee, 2016)." *(P7 L30ff.)*. Since we describe the data in the same subsection, we decided to describe ERA5 in the same subsection and not in a separate one.

**R**: Section 2.2.7, please update your reference to Patilea (now published).
**A**: Done *(P8 L21)*.

**R**: 2.2.8 NOASIM: "with the help of an genetic algorithm" change to "a genetic"
**A**: Done *(P9 L29)*.

**R**: Please adopt a more descriptive title for section 4.
**A**: We changed the title to "Sea Ice Concentration" *(P12 L4)*.

**R**: Page 10, line 20. If you use OSI-450, the correct reference is Lavergne et al. 2019.
**A**: The reference was changed to Lavergne et al. (2019) *(P12 L10)*.

**R**: Line 21: there are only 30 days in April.
**A**: Corrected, thanks *(P12 L12)*.

**R**: One would expect a nearly 100% average SIC north for Greenland in winter. So it is either an artefact of the OSI SAF data, or of your box that is too large and includes the East Greenland sea region. Discuss.
**A**: The magnitude of the OSI-SAF SIC did not change much after adapting the box of Moore et al. (2018). We visually inspected the time series of average SIC for each year of the climatology. We found that the mean was seldom close to $100\,\%$. The years before

1987 had a slightly lower mean than the years after 1987. Before 1987, SMMR data were used, while SSM/I data were used after 1987, so that there may be a problem with the SMMR data. However, years with mean SIC close to 90 % also occurred after 1987, so that this can not be the only reason. We conclude that this is a shortcoming of the OSI-SAF SIC, but does not contradict our statement that the SIC of this year was very significantly lower than normal.

**R**:Also here: you stitched together two SIC products (OSI-450 and OSI-401) and should first comment the temporal consistency of the two, before comparing winter 2017/18 to the climatology.
**A**: Done *(P12 L11)*.

**R**: Avoid "this year's event" and rather refer to the 2017/18 winter season explicitly.
**A**: Done *(P12 L16)*.

**R**: Figure 3 : in what respect is your merged MODIS+AMSR2 SIC much better than the OSISAF SIC (Figure 1)? The OSI-SAF curve on Figure 1 also reaches 0.7 mean SIC. Maybe Mean SIC is not the most appropriate metric here, did you try "open water extent" (1 – sea-ice-extent)? And -again- your box seems very large wrt to the polynya (extending to the East Greenland Sea). Bring the OSI SAF curve from Figure 1 onto Figure 3, and discuss.
**A**: We have calculated the open water extent for the merged SIC and the AMSR2 SIC and included in a Figure of its own (Figure 2 in this document). The same box as in Moore et al. (2018) was used. We constrain our comparison to cloud-free pixels and to scenes where at least half of the box was covered. The open water extent is normalised to the number of cloud-free pixels. The time series in Figure 2 shows that the absolute differences are between 1 and 3 %. As the benefit of the merged product's higher resolution lies mainly in the retrieval of the leads which form when the polynya starts to open, we also show the quotient of the merged and the AMSR2 SIC during the opening phase of the polynya (lower panel in Figure 2). Here, we see that the merged product is able to retrieve up to 60 % more open water than the AMSR2 SIC alone. This reflects the leads which the AMSR2 SIC do not retrieve.
It is hard to compare this to the OSI-SAF SIC for two reasons. The first one is the low bias of the OSI-SAF SIC which we mentioned above. This makes it more likely for OSI-SAF SIC to have a high open water extent despite its coarse resolution. The second one is that we would need to compare daily means (OSI-SAF SIC) to single overflights (merged SIC). Since we want to focus on the benefit introduced by the merged product's higher resolution, we decided to not include the OSI-SAF SIC and hope that you agree with us in this point. The calculation of the open water extent is described in Section 3.3 *(P11 L15ff.)*, the results are described on *(P13 L31 ff.)*.

[Figure]

Figure 2: Time series of the merged and AMSR2 open water fraction. The upper panel shows the absolute value of the open water fraction for both datasets. The middle panel shows the difference of the merged and the AMSR2 open water fraction. The lower panel shows the quotient of the two datasets during the opening phase. Mind the different time span of the lower panel. Only points where MODIS data were available were considered.

**R**: Page 15, line 11-13: "Note that the area of the opening is still visible as dark/new ice in the Sentinel-1 mosaic. This shows the limitations of AMSR2 data for the observation of polynya events". I do not understand. If a uniform cover of new-ice is seen in the Sentinel-1 mosaic, then it is correct for AMSR2 SIC to show 100% SIC (irrespective of its thickness if > 10cm). If one is interested in SIC, the SAR image might be hard to turn into 100% SIC, so I would see this as a limitation of the SAR technique. Explain further what you mean, or remove.

**A**: What we meant was that the SAR data allow identification of the polynya region even after the polynya is refrozen, while the AMSR2 SIC do not. This is, however, not the focus of our paper, so we decided to remove this sentence *(P18 L12f.)*.

**R**: Figure 4: lower panel: it would be good to align the design (e.g. yticklabels, gridlines, with that of Figure 5).

**A**: Done. The only remaining differences are that Figure 4 shows a time series of all merged overflights while Figure 5 shows daily means. Also, the acquisition times of the four maps shown in Figure 4 are marked in Figure 4 i). The next Figure shows the new version of Figure 4i (upper panel) and Figure 5 for comparison.

[Figure]

(a) New version of Figure 4i: Time series of the polynya area. The polynya area is calculated as sum of the open water fraction (1 - merged SIC) in the map area, multiplied with the respective grid cell size. All available granules are shown. The acquisition times of the 4 maps shown in the paper are marked by the vertical dashed lines.

[Figure]

(b) Lower panel of Figure 5: Same time series as in upper panel, but with daily means instead of all granules.

Figure 3: New version of polynya area time series: The layout of Figure 4i (upper panel here) has been adapted to match that of Figure 5 (lower panel here).

**R**: You could consider merging together section 4 and 5 because section 4 is quite short.

**A**: If the reviewer agrees, we would like to keep the sections separated. We like the structure of having one section (section 4) which presents and discusses the SIC products and the polynya itself, one section (section 5) which explains how the polynya opened and one section (section 6) which focuses on the sea ice production in the polynya.

**R**: Section 5: "Given the time of year and the location, only anomalous sea ice drift can be the driver. Nevertheless, a warm-air intrusion (Fig. 5 and 6) contributed to maintaining the polynya open." here you are stating the conclusions of your analysis. Move them to the end of this section, after the presentation of the results.

**A**: A similar statement is already made at the end of the section *(P20 L27ff.)*, so we removed the sentences you named *(P20 L3ff.)*.

**R**: "We conclude that the sea ice must have been broken up by sea ice drift." Since Moore et al (2018) exists, you should acknowledge them here, e.g. by stating that you confirm (or not) their conclusions.

**A**: We have added to Section 7 (Discussion) that our findings are consistent with those by Moore et al. (2018) *(P30 L10ff.)*. They find that the sea ice thickness loss due to sea ice motion is much higher than the thermodynamical sea ice growth. Also, their time series of ice loss due to ice motion is consistent with our drift time series.

**R**: Figure 5: fix unit of y-axis for lower panel (should be km^2)

**A**: Fixed.

**R**: Figure 6 and 7 (d) and (e): please regrid the winds to a polar grid before plotting. Here the lat/lon original sampling is evident and disturb the interpretation (e.g. no vectors in the central Arctic Ocean).

**A**: We have regridded the winds.

**R**: Table 1 and Figure 8: consider re-stating the source of sea-ice drift information (OSI SAF) in the legend, like done in Figure 6 and 7 (ERA5).

**A**: Done.

**R**: Section 6: We need more details on how the NAOSIM, FDD, and SMOS/SMAP products are compared. The sentence "For consistency, only grid cells with a SIC minimum beneath 50% during the polynya on the respective grid are considered for the calculation. " in the caption to Figure 9 should be re-written (I did not understand what you mean), and this should be in the text of section 6 (not in the caption).

**A**: When comparing the modelled (NAOSIM, FDD) to the observed (SMOS/SMAP) values, we wanted to avoid interpolation from the model grid to the observation grid. We therefore defined the polynya area on each grid as the area of all pixels which had less than 50 % at least once during the polynya period. The SIC from the NAOSIM model was used for the determination of the criterion *(P9 L1ff.)*. For deriving the polynya area of the FDD thermodynamical growth, we used the ERA5 SIC *(P8 L17)*. For deriving the area of the SMOS/SMAP sea ice thickness, we used ASI-AMSR2 values at 12.5 km grid spacing *(P8 L26f.)*.

**R**: The SMOS/SMAP curve is unsettling because it first drops while the two others grow steadily. Yet, you write "The SMOS/SMAP sea ice thickness evolved synchronously to the accumulated thermodynamic sea ice growth" which is maybe correct after the re-freeze has started, but not before. We need more details on how you extracted the SMOS/SMAP curve (average over the same box)? What you probably want is to show SMOS/SMAP only there there is newly forming ice (not where there is still old sea-ice).

**A**: Based on the ASI-AMSR2 SIC at 12.5 km grid spacing, we derive a mask of pixels

which had less than 50 % SIC at least once during the polynya event *(P8 L26f.)*. This is the same procedure as described in our answer above. Meaningful comparison between the SMOS/SMAP sea ice thickness and the NAOSIM and FDD sea ice thickness is only possible after the refreezing has started and before the sea ice thickness exceeds 50 cm. We therefore plotted the SMOS/SMAP sea ice thickness in a light shade and dashed during the opening phase of the polynya and after it reached 50 cm.

**R**: Also, the SMOS/SMAP algorithm uses the hypothesis of 100% sea-ice cover (this is not the case during the whole event).
**A**: Especially for very low sea ice thickness, it is not possible to disentangle sea ice concentration and sea ice thickness. This decreases the reliability of the SMOS/SMAP sea ice thickness during the very early freeze-up. It is also partly responsible for the decrease of the SMOS/SMAP sea ice thickness while the polynya breaks up. This has been added to the paper *(P24 L23ff.; P31 L2ff)*.

**R**: And L-band radiometry might (or not, discuss) be affected by the warm air intrusion if there is surface melt.
**A**: The warm air intrusion happened mainly when the polynya was opened. Even then, surface melt likely only occurred sporadically because the 2 m air temperature was only above the freezing point occasionally (Fig. 5, upper panel). The period which is relevant for the comparison of the thermodynamical growth starts on February 25th. At this time, 2 m air temperatures were already -10/-15 °C (daily maximum/mean). Afterwards, it got even colder. We conclude that surface melt did not influence the SMOS/SMAP sea ice thickness retrieval during the phase which we focus on. This has been added *(P31 L15ff.)*.

**R**: Please re-work this section.
**A**: The section has been rewritten as described above. We hope that it is better understandable now.

**R**: Sea-ice volume computations and Fig 9 (b): first, methodology: "by multiplying the accumulated growth rates from Fig. 9 with the area covered by the polynya": where do you find the area of the polynya? Is it from Figure 4 and 5 (lower panels)? If the case, then after March 8th the area is 0 (according to the merged product), so we would expect the volume curves to not grow anymore?
**A**: We have used the same area which has been used for the sea ice thickness time series in Figure 9. It was derived as the area of all pixels which have been beneath 50 % SIC at least once during the polynya event. Because this is a fixed area, the volume curves increase until the end of March.

**R**: Looking again at Figure 9 (b) it almost seems it is a scaling from Figure 9 (b). So did you use a fixed polynya area? If a fixed polynya area, the plot does not bring information compared to your sentence "The freezing degree day parameterisation yields a sea ice volume of 33 km3 , NAOSIM yields a sea ice volume of 15 km3 ." Please rework the description of your "new ice volume" computation.

**A**: We did indeed use a fixed polynya area, therefore it is a scaling from Figure 9 (a) (we assume that you meant Figure 9 (a) and not Figure 9 (b)). We agree that this plot is not necessary and remove it.

**R**: Page 23 line 6: This would be Figure 9(c).
**A**: Thanks for pointing this out.

**R**: Figure 10: panels to the right with AEM. Interesting, but difficult to know where we are. Suggestion to plot a larger area or add a larger map as inlet to show where the sampled region is.
**A**: We have reshaped the plot and added an inset to Fig. 10a):

[Figure]

Figure 10: AEM measurements of the ice thickness of young first year ice formed in the polynya until the end of March. (a): Normalised histogram of the AEM measurements. The shades indicate the standard deviation of the three single flights. The vertical lines show the mean sea ice thickness calculated based on the freezing degree days (dashed) and the NAOSIM model (dotted) since February 14th. The dotted polygon shows the region of the polynya. The small black rectangle shows the region of the AEM flights. (b) and (c): AEM flights on March 30th and 31st. The dots show the AEM measurements, averaged every 5 km. The background shows a Sentinel-1 mosaic for the respective day.

**R**: Figure 10: Also, merge the two maps and use a different symbol (or a label) to show the difference of date.
**A**: If the reviewer agrees, we would like to keep both maps. The reasons is that they show S1 mosaics for the respective date and it seems more consistent to us to show the mosaic for the corresponding day for each flight.

**R**: Figure 10: The AEM frequency distribution has a first peak around 0.1m. What is this, an artefact? Describe and comment.
**A**: The peak is not an artefact. It is indicative of a small number of leads and very thin ice that was identified by visual inspection. We added the sentences " The small mode is caused by leads which refreeze rapidly to form dark or light nilas. This explains the presence of classes of very thin ice adjacent to the open leads." *(P26 L13ff.)*

**R**: Section 7, discussions: "extraordinary" is often (always?) used in a very positive sense, while you mean here this is a first-time event.
**A**: "an extraordinary event" has been replaced by "a first-time event" *(P28 L6)*.

**R**: Again, Moore et al. (2018) already compared with the 1978-2017 climatology, so your should put your findings in perspective of their study.
**A**: We have added to our paper that we confirm and extend the results of Moore et al. (2018). While they show that the mean February SIC in 2018 was lower than any mean February SIC from 1978-2018, we show that this statement holds for the entire period from October 1st to April 30th. *(P28 L4ff.)*

**R**: Page 29, line 30: "European Copernicus Sentinel-2" add "Union".
**A**: Done *(P33 L23ff.)*.

**R**: Acknowledgements: since you use both SIC and SIDrift from EUMETSAT OSI SAF, you could add them in this section. Also, you could credit DMI for running (and sharing data from) the weather station.
**A**: Done *(P35 L2ff.)*.

We hope that we have addressed your comments in a satisfying manner and thank you again for the time you took to assess our manuscript.

▷ Subsections 2.1.1 (AMSR2 sea ice concentration) and 2.1.2 (MODIS data) now contain more information about the orbit characteristics of GCOM-W1, Aqua and Terra.

▷ The description of how the airborne ice thickness profiles have been moved from the section 2 to section 3.

▷ More detailed information about how the sea ice thickness from the NAOSIM mode, the freezing degree day parameterisation and the passive microwave SMOS/SMAP product are compared have been included into the respective subsections in section 2.

▷ Section 3 now includes the calculation of the Open Water Extent.

▷ The title of section 4 has been changed from "Polynya" to "Sea ice concentration".

▷ A description of the spatial standard deviation of SIC in the polynya region (Fig. 1, lower panel) has been added at the end of subsection 4.1.

▷ A Figure with time series of the open water extent of the merged SIC and the AMSR2 SIC has been included (Figure 4). It is described at the end of subsection 4.2.

▷ A comparison of our results to those of Moore et al. (2018) has been added in subsections 5.1 and 7.

▷ In section 6, a short description of how the different datasets are compared has been included, together with a reference to the more detailed description in section 2.

▷ In section 6, we discuss whether and how the SMOS/SMAP sea ice thickness was influenced by the fact that SIC was beneath 100% during the opening of the polynya.

▷ In section 6, we added a more detailed description of the frequency distribution of the AEM sea ice thickness.

▷ In Figure 10, we removed the middle panel which contained a time series of the sea ice volume produced thermodynamically in the polynya.

▷ Figure 11 has been reshaped, a map has been included to show the region of the AEM flights.

▷ The advantage of the higher resolution of the merged SIC towards the AMSR2 SIC is discussed in section 7, based on the time series of open water extent in Figure 4.

▷ We compare our results of sea ice drift in the polynya region to those of Moore et al. (2018).

▷ The evolution of the sea ice thickness is discussed in more detail in section 7.

▷ A more extensive conclusion regarding the sea ice growth in the polynya has been included in section 8.

**References**

[revised manuscript text omitted]

---

## Referee Report (RR1)

**Review of revised version of Ludwig et al. "The 2018 North Greenland polynya observed by a newly introduced merged optical and passive microwave sea ice concentration dataset".**

I would like to thank the authors for their thorough revision work and the answers they provided to my earlier comments and reviews. The manuscript is now much better balanced between the introduction of a new AMSR2+MODIS merged SIC product, and the discussion of the specific Feb 2018 polynya. The polynya part still stands stronger than the merging part, but this is much improved with respect to the first version.

I can recommend the publication of this manuscript in The Cryosphere. I provide below some comments and suggestions that I hope the authors can consider while working on their final revision. I also report some typos.

**Comments and suggestions for further analysis**

The manuscript is quite thorough (not lengthy) in its analysis of the polynya event. I would however suggest to put slightly more efforts on the climatological context (Figure 1).

The first suggestion is to move from using OSI-401 (the operational SIC product of OSI SAF) to using OSI-430-b (the newly released Interim Climate Data Record – ICDR). OSI-430-b is built with a main objective to be consistent with OSI-450 from January 2016 onwards. Since OSI-430-b has a 16 days latency, it fully covers your time period. You can find OSI-430-b information and data from http://osisaf.met.no/p/ice/ice\_conc\_cdr\_v2.html. It should not be much additional work, and would save you from commenting the consistency between OSI-450 and OSI-401 in section 2.2.1 and 4.1. On that topic, the sentence "The time series of both products is consistent at the transition (Lavergne et al. 2019)" (p. 6 lines 12-13) is very far fetched, since Lavergne et al al. 2019 does not deal with neither OSI-401, nor OSI-430-b. The consistency of OSI-450 and OSI-430-b at the transition is -at time of writing- only documented in the Validation Report for the OSI-450 and OSI-450 and

The second suggestion for Figure 1 starts as a question: is the 1979-2017 average computed from the "ice\_conc" variables in OSI-450 data files, or does it also use "raw\_ice\_conc\_values"? Variable "ice\_conc" only contains SICs up to 100%, while "raw\_ice\_conc\_values" gives access to the full error distribution of retrieved SICs, that is including values retrieved above 100%. A suggestion is thus to combine variables "ice\_conc" and "raw\_ice\_conc\_values" to prepare a non-thresholded SIC field before averaging. Using the non-thresholded is the correct way to prepare Figure 1, as otherwise the average curve can only be below 100%. The impact on the curve is not known, but it might raise the mean level to something closer to 100% (as expected). If you implement this change, please add a sentence to describe that you did combine the two variables before averaging. If you do not implement this change, please add a sentence stating that your average did use the thresholded SICs, and thus must result in lower SICs than could have been the case.

Finally, I would invite you to include in your discussion a dedicated paragraph on the difference between lead fraction products, and SIC products. You have elements of this in your text, but it would be an added value of this paper if you could wrap this in a "discussion" paragraph. In effect, you consider your merged SIC product is better because it shows more leads (at the beginning of the event). However, as you note, these leads might well be already refrozen when observed by MODIS (and AMSR2). In that case, a perfect SIC product should not show them at all. I understand the thrust towards showing high-resolution features such as leads in coarser resolution products, but should they all really be shown as reduced sea-ice concentration? Are there other ways the lead information could be shipped to users? Please discuss shortly.

**Minor edits and typos**

Throughout the text. It is a widely accepted convention to use "sea ice" as a noun ("sea ice froze rapidly"), but "sea-ice" when qualifying one of its characteristics ("the sea-ice concentration dropped", "sea-ice edge moved …"). Consider fixing throughout the text.

All the maps: you consistently have your Greenland filled with green (!). However many of your colormaps contain the color green (e.g. Figure 5, Figure 9). Consider finding a better color for filling Greenland.

P. 1, line 14 : "Two estimates of thermodynamic ... of 60 and 65 cm at the end of March in the area opened by the polynya".

P2. line 3: "the fraction of a given ocean area which..."

P2. line 16: "from 40-50 km from 19 and 37 GHz channels...."

P2. line 19: "prevents accurate monitoring of the sea ice edge" (or any better word that does not repeat resolution/resolve).

P2. line 21: replace "which are typically" by "and are typically".

P3. line 11: "The benefit of merged SIC with respect to single-sensor..." (or compared to).

P3. line 12: "during the formation of a polynya..."

P3. line 24: "Single latent heat"... what does "single" means here. Remove or clarify.

P5. line 5: "Sect. 9 lists..." or "provides"...

P5. line 15: Find a more explicit heading. Maybe "Input SIC data to merged product"?

P5. line 17: "It's orbit has an inclination..."

P6. section 2.2.1: specify that the OSI SAF SIC OSI-450 and OSI-430-b uses "coarse" resolution instruments (SMMR, SSM/I, SSMIS).

P6. line 19: you so far only introduced SIC, so it is not optimal to list new atmosphere and drift variables in the opening of this section.

P7. eq for FDD: please use index "n" inside the sum  $(T^air_{n})$ .

P8. line 9: what "passive microwave sea-ice concentration" must be beneath 50%? ASI? Please clarify.

P8. section on the NOASIM model. This section is much more detailed than the other. It can be justified because it is the only model, but please consider if this section can be shortened and the interested reader referred to publications about NAOSIM (e.g. the initialization of the model since 1980)?

P8 section on the NOASIM model: are the runs free runs or runs with data assimilation? Clarify, and shortly describe the assimilation (methodology, variables assimilated) if the model is not in free run.

P10, line 21: "80% of the picels were covered at least once". What does "covered" mean here? Cloud-covered? Covered by the product (hence cloud-free)? Clarify.

P10, line 25-26: move the discussion about the A-train and the Aqua/Terra platform to section 2.1.2.

P11, line 29: "visual observations showed"... where do these come from? During the flights? In-situ on sea-ice? Visual interpretation of satellite/aerial images? Please clarify.

P13, line 24: "The reason is again that they are more sensitive to newly formed sea ice"?

P13, line 28: define "open water extent". Does it use a 15% threshold?

P15, Figure 3: consider choosing another color scheme (blue and green and red can easily be mixed-up by color-blind persons).

P16, Figure 4: it is not obvious what the two lower panels bring. Would one of them (the difference) be enough? Or a third one: (Merged-AMSR2)/AMSR2?

P18, Figure 5 (right column). What is the unit? Could the colorscale be changed to show more contrast?

P21 replace "This year" with "During the polynya event,.."

P23: it is not clear if, e.g. +90, is westward or eastward drift.

P27, line 5-6 "Our statement is even stronger..." consider another formulation, e.g. "We confirm and further strengthen the findings of ..."

P28, line 14-15 "Also, refrozen leads with are covered by snow …" Exactly the type of argumentation that could enter a discussion paragraph about high-resolution features (refrozen leads) into sea-ice concentration products (as suggested in my comments earlier).

P29, line 29: remove the last sentence (or merged with that line 25).

P31, line 10: "SIC shows closed"

P31, line 27: "showed more SIC values between "

P32, line 6: "Events like this have occurred before ..." add "but not with the same magnitude" ?

---

## Author Response (AR2)

**Answers to Anonymous Referee 1**

We sincerely thank you for again taking your time to review our paper. Your comments greatly helped to improve structure and clarity of our manuscript. Please find our detailed response to your comments below. Your comments are marked by an **R**, our answers are marked by an **A**.

**R**: Page 2, line 22: The MODIS Terra product archive starts on 24 February 2000, not on the launch date in 1999. S-NPP VIIRS was launched 28 October 2011. But the first date of release of science data products is 19 January 2012. In this context the date of the first available products should be given, instead of the launch date. Also need to correct those dates in Sect. 2.1.2.
**A**: Thanks for this hint, we have adjusted the dates accordingly.

**R**: Check that paragraph indents are consistent. Page 3 has inconsistent paragraph indents.
**A**: We have made sure that the paragraph indents are consistent throughout the paper.

**R**: Page 5 line 5: Since the MODIS Aqua product was used it should be referred to as MYD29 throughout the manuscript to clearly identify it as the Aqua product.
**A**: We now refer explicitly to MYD29.

**R**: Page 5 line 10: I think I would be more clear to write "... and the data (Sect. 2.2) we used to investigate reasons..." rather than having (Sect. 2.2) at the end of the sentence.
**A**: Thanks, we have reshaped the sentence according to your suggestion.

**R**: Page 12 line 3: Perhaps restructure the sentence so it does not start with "Figure 2 shows". Awkward style to start a sentence with "Figure".
**A**: The sentence has been restructured and now reads "The AMSR2 SIC, MODIS SIC, merged SIC and SAR SIC are compared in Fig. 2a)–d)"

**R**: In the Outlook section, second sentence: "The benefit towards single-sensor SIC is demonstrated." Is that what you intended to say? Or, should "towards" be "over". Your merged SIC product is good and has advantages over single-sensor as stated in conclusions section, but this sentence does not support that.
**A**: Thanks for pointing this out, we have replaced "towards" by "over".

We hope that we have addressed your comments in a satisfying manner and thank you again for your assessment of our paper.

**Answers to Thomas Lavergne (Reviewer 2)**

Please have our sincere thanks for taking your time to assess our paper a second time and recommending publication in The Cryosphere. Your constructive and encouraging feedback is greatly appreciated. Please find our detailed response to your comments below. Your comments are marked by an **R**, our answers are marked by an **A**.

**R**: The first suggestion is to move from using OSI-401 (the operational SIC product of OSI SAF) to using OSI-430-b (the newly released Interim Climate Data Record – ICDR). OSI-430-b is built with a main objective to be consistent with OSI-450 from January 2016 onwards. Since OSI-430-b has a 16 days latency, it fully covers your time period. You can find OSI-430-b information and data from `http://osisaf.met.no/p/ice/ice_conc_cdr_v2.html`. It should not be much additional work, and would save you from commenting the consistency between OSI-450 and OSI-401 in section 2.2.1 and 4.1. On that topic, the sentence "The time series of both products is consistent at the transition (Lavergne et al. 2019)" (p. 6 lines 12-13) is very far fetched, since Lavergne et al al. 2019 does not deal with neither OSI-401, nor OSI-430-b. The consistency of OSI-450 and OSI-430-b at the transition is -at time of writing- only documented in the Validation Report for the OSI-450 and OSI-430-b products (https://osisaf.met.no/docs).

**A**: Thanks for this suggestion. We now use OSI-430-b instead of OSI-401. Sections 2.2.1 and 4.1 have been adjusted accordingly. Regarding the sentence "The time series of both products is consistent at the transition", we now refer to the product user manual at `http://osisaf.met.no/docs/osisaf_cdop3_ss2_pum_sea-ice-conc-climate-data-record_v2p0.pdf` instead of Lavergne et al. (2019).

**R**: The second suggestion for Figure 1 starts as a question: is the 1979-2017 average computed from the "ice_conc" variables in OSI-450 data files, or does it also use "raw_ice_conc_values"? Variable "ice_conc" only contains SICs up to 100%, while "raw_ice_conc_values" gives access to the full error distribution of retrieved SICs, that is including values retrieved above 100%. A suggestion is thus to combine variables "ice_conc" and "raw_ice_conc_values" to prepare a non-thresholded SIC field before averaging. Using the non-thresholded is the correct way to prepare Figure 1, as otherwise the average curve can only be below 100%. The impact on the curve is not known, but it might raise the mean level to something closer to 100% (as expected). If you implement this change, please add a sentence to describe that you did combine the two variables before averaging. If you do not implement this change, please add a sentence stating that your average did use the thresholded SICs, and thus must result in lower SICs than could have been the case.

**A**: We have used the "ice_conc" variable. Thanks for pointing out that this must result in SIC beneath 100 %. We have decided to keep the time series as it is now and added the sentence "The OSI-SAF SIC are capped at 100 %, thus the SIC average can only be

beneath 100 % and must result in lower SIC than might have been the case. " in Section 4.1.

**R**: Finally, I would invite you to include in your discussion a dedicated paragraph on the difference between lead fraction products, and SIC products. You have elements of this in your text, but it would be an added value of this paper if you could wrap this in a "discussion" paragraph. In effect, you consider your merged SIC product is better because it shows more leads (at the beginning of the event). However, as you note, these leads might well be already refrozen when observed by MODIS (and AMSR2). In that case, a perfect SIC product should not show them at all. I understand the thrust towards showing high-resolution features such as leads in coarser resolution products, but should they all really be shown as reduced sea-ice concentration? Are there other ways the lead information could be shipped to users? Please discuss shortly.

**A**: If a lead is covered by sea ice thinner than approximately 10 cm, it will likely show up as reduced sea-ice concentration, even in the AMSR2 SIC. Leads covered by more than approximately 10 cm of sea ice can be retrieved by our merged product because of the larger sensitivity of the MODIS SIC to sea-ice thickness. For this study, we have decided for including these reduced SIC to indicate the presence of leads. In future, we could include a flag in our product which shows whether the AMSR2 SIC at the corresponding SIC is 100 %. The user could then decide her/himself whether she/he would like to have the lead shown as reduced SIC or whether she/he prefers to consider the lead as 100 % sea-ice covered. In that case, the flag could be used to set the corresponding SIC to 100 %. A paragraph discussing this has been included in the Discussion section.

**R**: Throughout the text. It is a widely accepted convention to use "sea ice" as a noun ("sea ice froze rapidly"), but "sea-ice" when qualifying one of its characteristics ("the sea-ice concentration dropped", "sea-ice edge moved ..."). Consider fixing throughout the text.

**A**: Thanks for this hint, we have implemented it throughout the manuscript.

**R**: All the maps: you consistently have your Greenland filled with green (!). However many of your colormaps contain the color green (e.g. Figure 5, Figure 9). Consider finding a better color for filling Greenland.

**A**: Greenland is now filled with a light brown, a color which is not contained in the colormaps.

**R**: P. 1, line 14 : "Two estimates of thermodynamic ... of 60 and 65 cm at the end of March in the area opened by the polynya".

**A**: This has been done.

**R**: P2. line 3: "the fraction of a given ocean area which..."

**A**: The word "ocean" has been added.

**R**: P2. line 16: "from 40-50 km from 19 and 37 GHz channels...."
**A**:The word "from" has been added.

**R**: P2. line 19: "prevents accurate monitoring of the sea ice edge" (or any better word that does not repeat resolution/resolve).
**A**:We adapted the word "monitoring".

**R**: P2. line 21: replace "which are typically" by "and are typically".
**A**: Done.

**R**: P3. line 11: "The benefit of merged SIC with respect to single-sensor..." (or compared to).
**A**: We adapted the formulation "with respect to".

**R**: P3. line 12: "during the formation of a polynya..."
**A**: Adapted.

**R**: P3. line 24: "Single latent heat"... what does "single" means here. Remove or clarify.
**A**: We removed the word "single". The sentence has been rewritten to be clearer. It now reads "The ten major Arctic coastal polynyas produce roughly 130–840 $km^3$ of sea ice per year (Tamura and Ohshima, 2011).".

**R**: P5. line 5: "Sect. 9 lists..." or "provides"...
**A**: We have adapted the formulation "Sect. 9 lists...".

**R**: P5. line 15: Find a more explicit heading. Maybe "Input SIC data to merged product"?
**A**: Thanks for the suggestion, we now use the heading "Input data to merged product" since here we describe the MODIS ice surface temperature and not yet the MODIS SIC.

**R**: P5. line 17: "It's orbit has an inclination..."
**A**: Implemented.

**R**: P6. section 2.2.1: specify that the OSI SAF SIC OSI-450 and OSI-430-b uses "coarse" resolution instruments (SMMR, SSM/I, SSMIS).
**A**: We added the sentence "It uses the coarse-resolution instruments SMMR (Scanning Multi-channel Microwave Radiometer), SSM/I (Special Sensor Microwave/Imager) and SSMIS (Special Sensor Microwave Imager Sounder."

**R**: P7. eq for FDD: please use index "n" inside the sum (Tˆair_n?).
**A**: The equation now reads

$$FDD = \sum_{n=1}^{n=n_{days}} [-1 * T_n^{air}],\tag{1}$$

where $n$ is the index of the respective day, $n_{days}$ is the total number of days and $T_n^{air}$ is the daily mean air temperature in ° C of the respective day.

**R**: P8. line 9: what "passive microwave sea-ice concentration" must be beneath 50%? ASI? Please clarify.
**A**: Yes, ASI. This is specified now in the text.

**R**: P8. section on the NOASIM model. This section is much more detailed than the other. It can be justified because it is the only model, but please consider if this section can be shortened and the interested reader referred to publications about NAOSIM (e.g. the initialization of the model since 1980)?
**A**: We have shortened the section and included literature references for the interested reader where we left out details.

**R**: P8 section on the NOASIM model: are the runs free runs or runs with data assimilation? Clarify, and shortly describe the assimilation (methodology, variables assimilated) if the model is not in free run.
**A**: There has been no data assimilation. We now state this in the text.

**R**: P10, line 21: "80% of the picels were covered at least once". What does "covered" mean here? Cloud-covered? Covered by the product (hence cloud-free)? Clarify.
**A**: We mean cloud-free. We have replaced "covered" by "cloud-free".

**R**: P10, line 25-26: move the discussion about the A-train and the Aqua/Terra platform to section 2.1.2.
**A**: We have moved the corresponding sentences as suggested.

**R**: P11, line 29: "visual observations showed"... where do these come from? During the flights? In-situ on sea-ice? Visual interpretation of satellite/aerial images? Please clarify.
**A**:The visual observations were made during the flights, this is stated now.

**R**:P13, line 24: "The reason is again that they are more sensitive to newly formed sea ice"?
**A**:We have included "newly formed".

**R**: P13, line 28: define "open water extent". Does it use a 15% threshold?

**A**: Yes, it does use a 15 % threshold. This is now described here in an extra sentence. Section 3.3 gives more details about the open water extent.

**R**: P15, Figure 3: consider choosing another color scheme (blue and green and red can easily be mixed-up by color-blind persons).

**A**: Thanks for this hint. We have partly chosen different colors and used a color-blindness simulator to make sure that the color scheme is distinguishable for color-blind people. Other Figures which might cause problems for color-blind readers were also adapted.

**R**: P16, Figure 4: it is not obvious what the two lower panels bring. Would one of them (the difference) be enough? Or a third one: (Merged-AMSR2)/AMSR2?
**A**: In our opinion, the middle panel shows the difference more clearly than the upper panel alone. For the lower panel, we follow your suggestion and use the normalised difference (Merged-AMSR2)/AMSR2) of the quotient Merged/AMSR2. This supports our finding that the benefit of the merged SIC is more pronounced during the early break-up phase of the polynya.

**R**: P18, Figure 5 (right column). What is the unit? Could the colorscale be changed to show more contrast?
**A**: The unit is greyscale values. This is now stated in the caption. We have made the range of the colorbar smaller to show more contrast.

**R**: P21 replace "This year" with "During the polynya event,.."
**A**: Done.

**R**: P23: it is not clear if, e.g. +90, is westward or eastward drift.
**A**: +90 means westward drift. We have added this in the text.

**R**:P27, line 5-6 "Our statement is even stronger..." consider another formulation, e.g. "We confirm and further strengthen the findings of ..."
**A**: We have adopted the formulation as suggested by you.

**R**: P28, line 14-15 "Also, refrozen leads with are covered by snow ..." Exactly the type of argumentation that could enter a discussion paragraph about high-resolution features (refrozen leads) into sea-ice concentration products (as suggested in my comments earlier).
**A**: A paragraph discussing the incorporation of high-resolution features into SIC products has been added (see also our answer to your previous comment).

**R**: P29, line 29: remove the last sentence (or merged with that line 25).
**A**: We have removed the last sentence.

**R**: P31, line 10: "SIC show**s** closed"

**A**: Thanks for pointing out this typo, we have corrected it.

**R**: P31, line 27: "showed more SIC values between "

**A**: We have included the word "values".

**R**: P32, line 6: "Events like this have occurred before ..." add "but not with the same magnitude" ?

**A**: We have added "but not with the same magnitude".

We hope that we have addressed your comments in a satisfying manner and thank you again for the time you took to assess our manuscript.

[revised manuscript text omitted]